# Protein compactness and interaction valency define the architecture of a biomolecular condensate across scales

**Anton A Polyansky[1,2]\*[†], Laura D Gallego[1,3][†], Roman G Efremov[4], Alwin Köhler[1,3,5], Bojan Zagrovic[1,2]\***

[1]Max Perutz Labs, Vienna Biocenter Campus (VBC), Vienna, Austria; [2]University of Vienna, Center for Molecular Biology, Department of Structural and Computational Biology, Vienna, Austria; [3]Medical University of Vienna, Center for Medical Biochemistry, Vienna, Austria; [4]MM Shemyakin and Yu A Ovchinnikov Institute of Bioorganic Chemistry, Russian Academy of Sciences, Moscow, Russian Federation; [5]University of Vienna, Center for Molecular Biology, Department of Biochemistry and Cell Biology, Vienna, Austria

**\*For correspondence:**
anton.polyansky@univie.ac.at (AAP);
bojan.zagrovic@univie.ac.at (BZ)

[†]These authors contributed equally to this work

**Competing interest:** The authors declare that no competing interests exist.

**Abstract** Non-membrane-bound biomolecular condensates have been proposed to represent an important mode of subcellular organization in diverse biological settings. However, the fundamental principles governing the spatial organization and dynamics of condensates at the atomistic level remain unclear. The *Saccharomyces cerevisiae* Lge1 protein is required for histone H2B ubiquitination and its N-terminal intrinsically disordered fragment (Lge1$_{1-80}$) undergoes robust phase separation. This study connects single- and multi-chain all-atom molecular dynamics simulations of Lge1$_{1-80}$ with the in vitro behavior of Lge1$_{1-80}$ condensates. Analysis of modeled protein-protein interactions elucidates the key determinants of Lge1$_{1-80}$ condensate formation and links configurational entropy, valency, and compactness of proteins inside the condensates. A newly derived analytical formalism, related to colloid fractal cluster formation, describes condensate architecture across length scales as a function of protein valency and compactness. In particular, the formalism provides an atomistically resolved model of Lge1$_{1-80}$ condensates on the scale of hundreds of nanometers starting from individual protein conformers captured in simulations. The simulation-derived fractal dimensions of condensates of Lge1$_{1-80}$ and its mutants agree with their in vitro morphologies. The presented framework enables a multiscale description of biomolecular condensates and embeds their study in a wider context of colloid self-organization.

## Editor's evaluation

In this work, the authors introduce and develop upon a computational model to investigate and quantify the effect of protein conformations and valence of interaction sites as organizers of structure within biomolecular condensates. The authors integrate their findings with new and emerging concepts regarding the coupling between phase separation and percolation as a determinant of driving forces and internal organization of condensates. The key insight that emerges from the current work pertains to the structure that prevails across length scales.

## Introduction

Biomolecular condensates, such as P-bodies, nucleoli, and stress granules, are membrane-less structures that contribute to the compartmentalization of the cell interior (*Brangwynne et al., 2009*; *Feng*

*et al., 2019*; *Lafontaine et al., 2021*; *Mitrea and Kriwacki, 2016*; *Mittag and Pappu, 2022*; *Molliex et al., 2015*). They have been implicated in diverse biological functions including transcription, signaling, and ribosome biogenesis (*Banani et al., 2017*; *Boeynaems et al., 2018*) and have also been linked with different pathologies (*Alberti and Dormann, 2019*). Importantly, major efforts have been invested in understanding the general physicochemical principles behind the formation of biomolecular condensates (*Alberti and Hyman, 2021*; *Banani et al., 2017*; *Brady et al., 2017*; *Brangwynne et al., 2015*; *Bremer et al., 2022*; *Dignon et al., 2018*; *Martin et al., 2020*; *Mitrea and Kriwacki, 2016*; *Pappu et al., 2023*; *Zeng et al., 2022*). While an early paradigm for understanding condensate formation has been liquid-liquid phase separation, mounting evidence suggests that a close coupling between segregative phase separation and associative network transition, that is, percolation, may be important in many cases (*Choi et al., 2020a*; *Choi et al., 2020b*; *Mittag and Pappu, 2022*; *Pappu et al., 2023*; *Schmit et al., 2020*; *Seim et al., 2022*).

Formation of biomolecular condensates has been observed for proteins (*Nott et al., 2015*; *Wang et al., 2018*; *Wei et al., 2017*), DNA (*King and Shakya, 2021*), RNA (*Jain and Vale, 2017*), and their mixtures (*Garcia-Jove Navarro et al., 2019*). Intrinsically disordered proteins (IDPs), in particular, feature extensively in many condensates and are thought to contribute to their formation via transient intermolecular contacts (*Banani et al., 2017*; *Uversky, 2021*). An important challenge in this regard has been to provide a multiscale picture of the spatial organization and dynamics of IDP condensates, connecting the conformational properties and interaction patterns of individual polypeptides in a crowded environment with the features of the condensates they build. Considering that IDP condensates are extremely dynamic and structurally heterogeneous, it is clear that such a picture needs to capture the statistical, ensemble-level aspects of how matter inside the condensates is organized.

While an atomistic view of individual proteins in crowded environments is currently beyond the reach of high-resolution experimental techniques, molecular dynamics (MD) simulations have been developed with precisely this aim in mind (*Dror et al., 2012*). Although limited in terms of sampling efficiency as compared to coarse-grained approaches (*Benayad et al., 2021*; *Dignon et al., 2018*; *Martin et al., 2020*), atomistic MD simulations can provide an accurate picture of protein structure, dynamics, and interactions with sub-Ångstrom resolution. For example, atomistic simulations have been used to model the dynamics of different peptides such as elastin-like peptide (*Rauscher and Pomès, 2017*) or different fragments of NDDX4 (*Paloni et al., 2020*) in condensate environment. Such simulations have also been combined with coarse-grained approaches and experiments (*Zheng et al., 2020*) and have provided a detailed view of the key interactions behind protein condensate formation (*Conicella et al., 2020*; *Murthy et al., 2019*; *Ryan et al., 2018*). Despite these important advances, however, the study of biomolecular condensates at the atomistic resolution is still at its beginning and many questions related to their most generalizable features are only starting to be addressed (*Conicella et al., 2020*; *Li et al., 2022*; *Murthy et al., 2019*; *Ryan et al., 2018*). These, in particular, concern the complex interplay between structure, dynamics, and thermodynamics of biomolecules in crowded environments.

In general, the concepts and methods of polymer physics have been widely used to understand the formation of biomolecular condensates (*Alberti and Hyman, 2021*; *Brady et al., 2017*; *Brangwynne et al., 2015*; *Dignon et al., 2018*; *Dzuricky et al., 2020*; *Guillén-Boixet et al., 2020*; *Martin et al., 2020*; *Mitrea and Kriwacki, 2016*; *Pappu et al., 2023*). However, in contrast to the dense, continuous phase observed upon macroscopic phase separation in simple polymers, many protein condensates tend to have a lower density and be enriched in water (*Alberti and Hyman, 2021*; *Keating and Pappu, 2021*; *Wei et al., 2017*; *Zaslavsky and Uversky, 2018*), making them closer in organization to typical colloids (*Slomkowski et al., 2011*). According to the definition used in colloidal chemistry, biological condensates share features with weak gels, which undergo a transition between a population of finite-size pre-percolation clusters or sol, and an infinitely large cluster or gel (*Stauffer et al., 1982*). In the case of biological condensates, phase separation coupled to percolation results in finite-sized colloidal clusters and the appearance of surface tension. In particular, under the requirement that the saturation concentration ($c_{sat}$) for phase separation is lower than the percolation concentration ($c_{perc}$), phase separation leads to an increase in local protein concentration and defines the phase boundary, while a percolation transition establishes network connectivity (*Mittag and Pappu, 2022*).

Importantly, starting with the seminal work of *Forrest and Witten, 1979*, it has been recognized that aggregates or clusters of colloidal particles typically exhibit fractal scaling, that is that the cluster

mass scales with cluster size according to a non-integer power law with the so-called fractal dimension $d_f$ as the exponent (*Lazzari et al., 2016*). In contrast to the regular geometric fractals, colloids generally build *statistical fractals* in which scaling laws hold between *average* values of mass and cluster size (*Havlin and Ben-Avraham, 1987*; *Stanley, 1984*). Overall, statistical fractal properties have been demonstrated under different conditions for many non-biological colloidal systems including silica, polystyrene, and gold colloids (*Lazzari et al., 2016*). Moreover, by using scattering techniques such as static and dynamic light scattering (*Lazzari et al., 2016*; *Lin et al., 1989*), or confocal and scanning electron microscopy (*Khatun et al., 2020*), the fractal nature has been associated with several biological colloidal systems, including different protein fibrils (*Knowles et al., 2007*; *Nicoud et al., 2015*), whey protein isolates (*Kharlamova et al., 2020*), and clusters of lysozyme (*da Silva and Arêas, 2005*) and amylin (*Khatun et al., 2020*). Finally, fractal model has recently been used to interpret the results of coarse-grained simulations and characterize the aggregation of a phase-separating intrinsically disordered huntingtin fragment (*Ruff et al., 2014*).

Following the paradigm set by fractal colloidal systems (*Lazzari et al., 2016*), we derive here a general analytical framework for modeling statistical fractal cluster formation involving biomolecules. As a key result, we connect the interaction valency and compactness of individual biomolecules in condensates with the fractal dimension $d_f$, which in turn enables us to describe the structural organization of the condensate at an arbitrarily chosen length scale. We apply the above framework to the N-terminal 80-residue fragment of Lge1, a scaffolding protein required for histone H2B ubiquitination during transcription (*Gallego et al., 2020*). $Lge1_{1-80}$ exhibits a strong compositional bias shared by many known phase-separating proteins (enriched in Y, R, and G, *Figure 1A*), is fully disordered, and undergoes phase separation readily (*Gallego et al., 2020*), making it a powerful system to study the general features of condensate formation in IDPs. We combine a detailed atomistic characterization of a model condensate containing 24 copies of $Lge1_{1-80}$, obtained via microsecond MD simulations, with our newly developed fractal formalism in order to propose an atomistic model of the $Lge1_{1-80}$ condensate extending to micrometer scale. To probe sequence determinants of $Lge1_{1-80}$ LLPS, we investigate its all-R-to-K (R>K) mutant, where the ability to form condensates in vitro is modulated, and its all-Y-to-A (Y>A) mutant, which impairs phase separation in vitro (*Gallego et al., 2020*; *Figure 1B*). Finally, we compare our theoretical predictions against the phase behavior of these different systems as studied experimentally via light microscopy. Our analysis provides a detailed, multiscale model of a biologically relevant condensate and establish a general, experimentally stestable framework for studying other similar systems.

## Results

### $Lge1_{1-80}$ condensate formation critically depends on tyrosine residues

We have first experimentally explored the solubility diagrams of WT $Lge1_{1-80}$ and its R>K and Y>A mutants by light microscopy using recombinant $Lge1_{1-80}$ peptides (*Figure 1—figure supplement 1A*). WT $Lge1_{1-80}$ phase separates already at a protein concentration of 1 μM in 0.1 M NaCl (*Figure 1B and C*). Its phase separation is robust and can be inhibited at high salt concentration (>2 M NaCl). The presence of tyrosine amino acids, which could provide competing π-π interactions, has no significant impact on WT condensate formation in the concentration range tested, while imidazole disrupts it at concentrations above 0.5 M. Mutating all arginine residues in WT $Lge1_{1-80}$ to lysine (R>K mutant) increases the $c_{sat}$ for phase separation by approximately 10-fold under the conditions tested (*Figure 1B and C*). The droplet-like WT and R>K condensates (*Figure 1B*, *Figure 1—figure supplement 1B*) display normal fusion behavior (*Videos 1 and 2*) and exhibit high circularity (*Figure 1—figure supplement 1C*). On the other hand, the substitution of the arginine guanidium groups, which are prone to π-π interactions, by the lysine amino groups, which lack sp2 electrons and do not participate in π-π interactions, makes the R>K mutant more sensitive to aromatic agents such as tyrosine (moderate effect) or imidazole (strong effect) (*Figure 1C*). At the same time, Y>A mutations strongly impair phase separation (*Figure 1B and C*, *Figure 1—figure supplement 1B*, see also *Gallego et al., 2020*), suggesting that tyrosines play a critical role in WT and R>K phase separation (*Figure 1C*) and highlighting the importance of π-π interactions in biomolecular condensate formation, as proposed earlier (*Vernon et al., 2018*). Moreover, the solubility diagrams are also in line with the observation that phase separation is related to the number of tyrosine and arginine residues in FUS and other

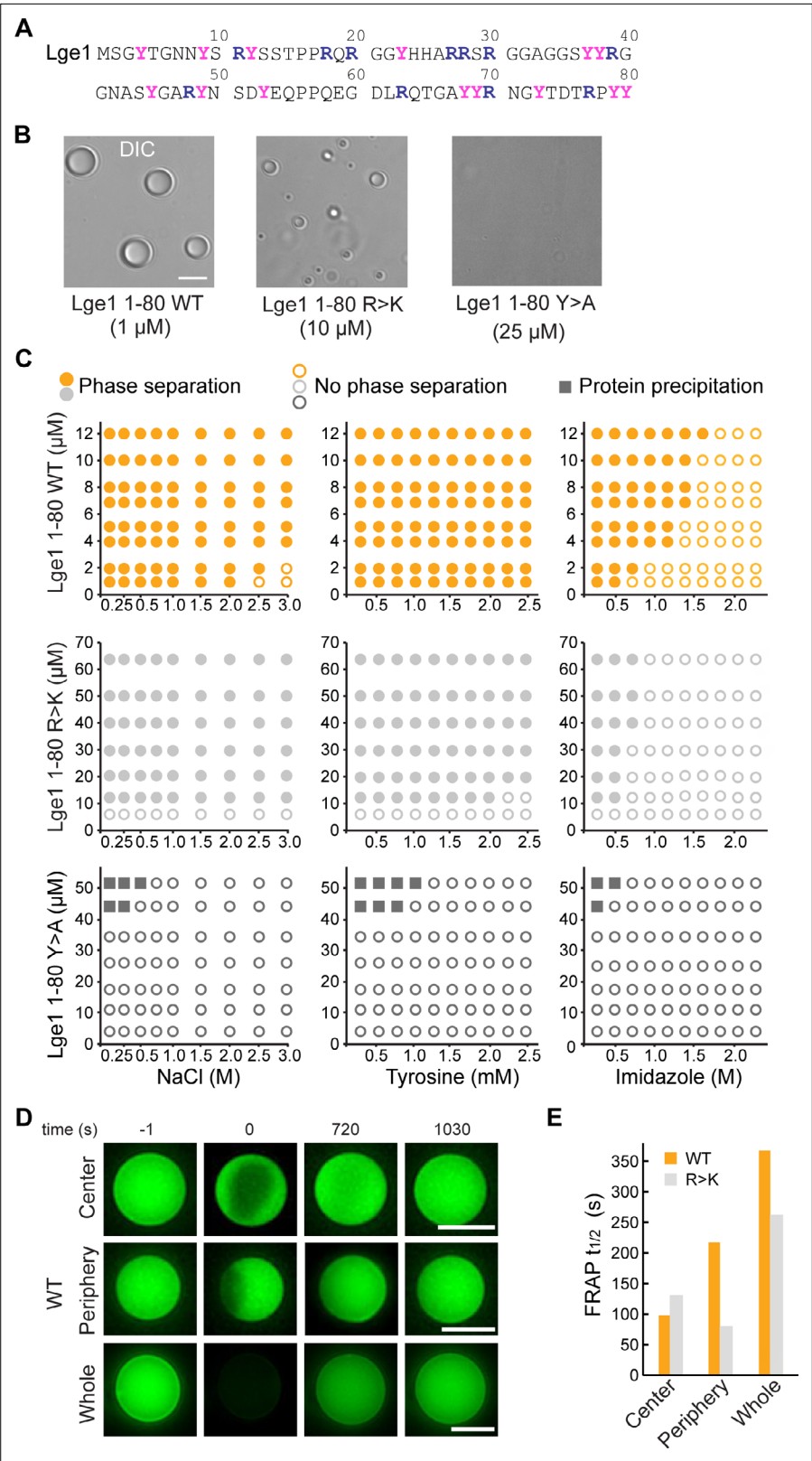

**Figure 1.** Lge1$_{1-80}$ condensate formation critically depends on tyrosine residues. (**A**) Sequence of Lge1$_{1-80}$. Arginines and tyrosines are highlighted in deep blue and magenta, respectively. (**B**) Condensate formation for Lge1$_{1-80}$ WT (left) and R>K (middle) in buffer with 200 mM NaCl. No such condensates are observed for Lge1$_{1-80}$ Y>A (right). Scale bar, 5 µm. (**C**) Solubility diagrams for the WT (*top row*), R>K (*middle row*) and Y>A (*bottom row*)

*Figure 1 continued on next page*

*Figure 1 continued*

Lge1₁₋₈₀ variants with protein concentration given on the *y*-axis and concentration of NaCl (*left panels*), tyrosine (*middle panels*), and imidazole (*right panels*) given on the *x*-axis. (**D**) Representative fluorescence recovery after photobleaching (FRAP) images of Lge1₁₋₈₀ WT condensates, bleached in the center (*upper panels*), periphery (*middle*), or across the whole condensate (*lower panels*), including pre-bleach (*left, time –1 s*), bleach (*time 0 s*), and post-bleach (*time 720 s, 1030 s*). Scale bars, 5 µm. (**E**) Half-times of FRAP of Dylight-labeled Lge1₁₋₈₀ WT (orange) and R>K (gray) that were bleached in the center, periphery, or across the whole condensate. Data was obtained after fitting to a double exponential model (see *Figure 1—figure supplement 2*).

The online version of this article includes the following source data and figure supplement(s) for figure 1:

**Source data 1.** Raw data used in panel C.

**Figure supplement 1.** Experimental and modeling studies of the effect of tyrosine and arginine mutations in Lge1₁₋₈₀.

**Figure supplement 1—source data 1.** Raw data used in panels A and C.

**Figure supplement 1—source data 2.** Calculated free-energy values used in panel E.

**Figure supplement 2.** Fluorescence recovery after photobleaching (FRAP) analyses of Lge1₁₋₈₀ and LAF-1 condensates.

**Figure supplement 2—source data 1.** Fluorescence recovery after photobleaching (FRAP) summary: fitting parameters, recovery half-times, and raw data.

phase-separating proteins (*Bremer et al., 2022*; *Dzuricky et al., 2020*; *Wang et al., 2018*). Finally, we note that at protein concentration of 45 µM and above, the Y>A mutant results in sporadic amorphous precipitates (*Figure 1C*).

We have used fluorescence recovery after photobleaching (FRAP) to characterize the overall dynamics of the condensates formed by WT and R>K Lge1₁₋₈₀ variants. First, the applied FRAP protocol was tested for LAF1 (see Methods) and a fluorescence recovery of up to 80% in approximately 20 min was demonstrated (*Figure 1—figure supplement 2H, I*), in agreement with the previously published data (*Taylor et al., 2019*). While the Y>A mutant was shown to be unable to form droplet-like clusters, the dynamic WT, and R>K condensates were studied using different bleaching strategies, including bleaching at the center, the periphery, or across the whole condensate (*Figure 1D*, *Figure 1—figure supplement 2A–G*). Calculation of the recovery half-times by using different single exponential models did not provide quality fits (*Figure 1—figure supplement 2H*, *Figure 1—figure supplement 2—source data 1*). Using double-exponent fitting allowed us to improve the quality of the fits and accurately describe the data (*Figure 1E*, *Figure 1—figure supplement 2A–C*, and *Figure 1—figure supplement 2—source data 1*). Thus, the WT and R>K recovery half-times are in the range of 100 s

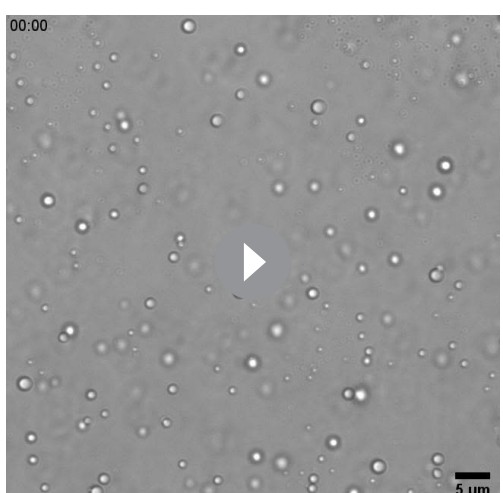 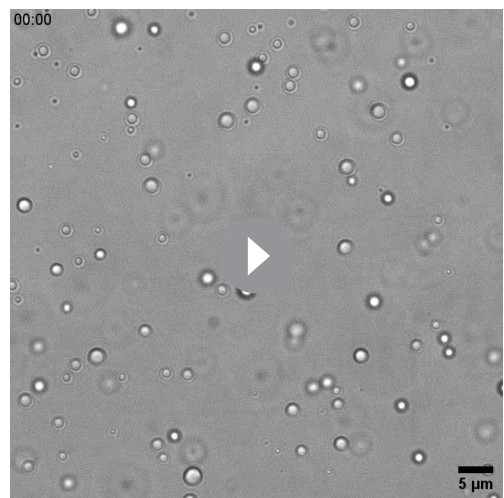

**Video 1.** Fusion of Lge1₁₋₈₀ WT condensates in solution. Protein concentration 1 µM. Scale bar, 5 µm.
https://elifesciences.org/articles/80038/figures#video1

**Video 2.** Fusion of Lge1₁₋₈₀ R>K condensates in solution. Protein concentration 10 µM. Scale bar, 5 µm.
https://elifesciences.org/articles/80038/figures#video2

for partial bleaching, and increase up to 350 s for bleaching of the whole condensates under the conditions tested. These values are similar to those previously reported for the in vitro condensates of other phase-separating proteins (*Lin et al., 2015*; *Taylor et al., 2019*). Importantly, the recovery half-time for the whole bleached WT condensates is approximately 30% higher than for R>K (*Figure 1E*), suggesting a potentially different internal organization of the two types of condensates.

To rationalize the biochemical effects of the mutations, we have calculated the interaction strengths of selected pairwise contacts (Y-Y, R-Y, K-Y). To this end, the binding free energies (ΔG) for these side-chain analogs pairs were determined using all-atom Monte-Carlo (MC) simulations in chloroform, methanol, DMSO, and water separately (*Figure 1—figure supplement 1D* and the corresponding source data). Interestingly, ΔG(Y-Y) is independent of the polarity of the environment (approx. –5 kcal/mol on average), while ΔG(R-Y) and ΔG(K-Y) depend significantly on the dielectric permittivity of the medium (*Figure 1—figure supplement 1D*), as shown for other aromatic sidechain analogs (F, W) (*Polyansky et al., 2009*) and nucleobases (*de Ruiter et al., 2017*) interacting with R and K. Overall, both ΔG(R-Y) and ΔG(K-Y) are lowest in the apolar environment (approx. –10 kcal/mol on average), intermediate in bulk water (approx. –7.5 kcal/mol on average), and highest at intermediate polarity, where they are comparable to ΔG(Y-Y). The strong dependence of ΔG(R-Y) on the properties of the environment and the fact that, regardless of the conditions, ΔG(Y-Y) is never more favorable than ΔG(R-Y) were also recently reported for simulations of complete amino acids (*Krainer et al., 2021*), although the exact values are difficult to compare due to differences in the exact nature of the simulated systems, the force fields used, and the method of how the ΔGs were derived.

In our simulations, significantly stronger R-Y binding as compared to K-Y was observed only in the environments of intermediate polarity (*Figure 1—figure supplement 1D*). This could contribute to the observed increase in the concentration required for Lge1$_{1-80}$ R>K phase separation as compared to the WT, especially if Lge1 condensates exhibit a lower dielectric constant than that of bulk water, as observed elsewhere (*Nott et al., 2015*). The effect of R>K substitutions on phase-separation behavior was also examined by others for different IDPs (*Bremer et al., 2022*; *Dzuricky et al., 2020*; *Schuster et al., 2020*; *Wang et al., 2018*) and was, furthermore, linked with the general differences in the intrinsic physicochemical properties of R and K (*Dubreuil et al., 2019*; *Fisher and Elbaum-Garfinkle, 2020*; *Fossat et al., 2021*; *Hong et al., 2022*; *Paloni et al., 2021*; *Zeng et al., 2022*). Our aim here is to explore it further in the context of Lge1$_{1-80}$ single- and multi-chain protein-protein interactions. Finally, the Y>A substitution likely causes a strong thermodynamic effect due to the removal of all possible R-Y and Y-Y intermolecular contacts.

## π-π interactions shape protein clustering in Lge1$_{1-80}$ condensates

To better understand the effect of mutations on the organization of Lge1 condensates, we have performed three 1-μs-long MD simulations with 24 copies each of Lge1$_{1-80}$ WT, Y>A, and R>K mutants in the mM concentration range as well as control simulations of the three proteins present as single copies (*Table 1* for details). In our simulations, proteins tend to form clusters that are characterized by pronounced structural heterogeneity and dynamics (*Figure 2A*; *Video 3*). Specifically, the WT 24-copy system forms a single percolating cluster over the last 0.3 μs (*Figure 2C*) with all copies of the protein engaged. On the other hand, the mutants do not form such an extensive interaction network, but rather associate in multiple, differently sized clusters. The large continuous WT cluster is shaped by π-π interactions (*Figure 2A and B*) between the most abundant amino acids (R, Y, G), whereby R-Y

**Table 1.** Details of simulated systems including composition, effective molar and mass protein concentration, size of the simulated cubic box, simulation time, and the number of replicas.

| Name | Protein | Water | Na+ | Cl- | [Protein], mM | [Protein], g/l | Box size, nm | MD time, μs | Replicas |
|---|---|---|---|---|---|---|---|---|---|
| Lge1 1–80 WT | 1 | 23674 | 44 | 50 | 2.3 | 20.7 | 9.0 | 1 | 2 |
| Lge1 1–80 Y>A | 1 | 23742 | 44 | 50 | 2.3 | 17.7 | 9.0 | 1 | 2 |
| Lge1 1–80 R>K | 1 | 23673 | 44 | 50 | 2.3 | 20.0 | 9.0 | 1 | 2 |
| Lge1 1–80 WT 24 copies | 24 | 182056 | 351 | 495 | 6.9 | 62.5 | 18.0 | 1 | 1 |
| Lge1 1–80 Y>A 24 copies | 24 | 183407 | 351 | 495 | 6.9 | 45.0 | 18.0 | 1 | 1 |
| Lge1 1–80 R>K 24 copies | 24 | 217857 | 413 | 557 | 5.8 | 50.8 | 19.0 | 1 | 1 |

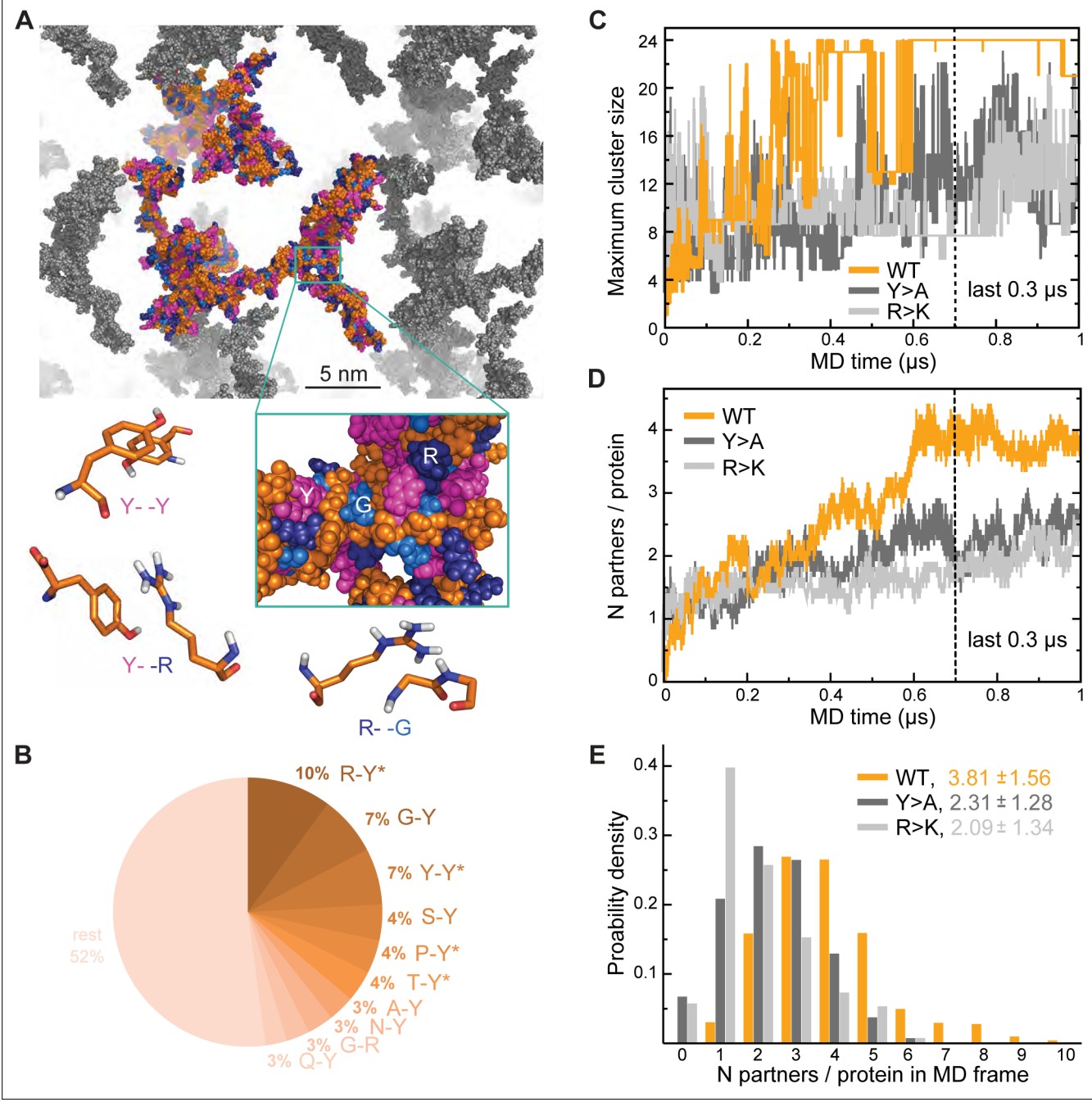

**Figure 2.** Analysis of interaction networks for Lge1$_{1-80}$ variants by all-atom molecular dynamics (MD) in 24-copy systems. (**A**) Exemplary MD snapshot of the WT interaction network (see *Video 3* for the full MD trajectory). Proteins in the simulation box are given in the atomic representation (orange), whereby glycine, arginine, and tyrosine residues are colored in sky blue, deep blue, and magenta, respectively. Periodic images of the simulated system are shown in gray. (**B**) Pairwise contact statistics over the last 0.3 μs of simulations for the Lge1$_{1-80}$ WT. Contacts enriched over the sequence background are marked with stars (see *Figure 2—figure supplement 1A* for exact enrichment values). (**C**) Time evolution of the largest detected protein cluster size and (**D**) the average number of interaction partners per protein chain, that is interaction valency, for the WT (*orange line*), Y>A (*dark gray line*), and R>K (*light gray line*) multi-chain systems. (**E**) Distributions of the number of interaction partners per protein over the last 0.3 μs of simulations. The color code is the same as in (**C** and **D**).

The online version of this article includes the following source data and figure supplement(s) for figure 2:

**Source data 1.** Contact statistics used in panel B.

**Source data 2.** Molecular dynamics (MD) data used in panels C, D, E.

**Figure supplement 1.** Inter- and intramolecular interactions of Lge1$_{1-80}$ variants in molecular dynamics (MD) simulations.

*Figure 2 continued on next page*

*Figure 2 continued*

**Figure supplement 1—source data 1.** Molecular dynamics (MD) data used in panels A, B, C.

contacts dominate (10% of all possible pairs, *Figure 2—figure supplement 1A*), followed closely by G-Y and Y-Y contacts (7% of all possible pairs each, *Figure 2—figure supplement 1B*). In particular, both R-Y and Y-Y contacts are strongly enriched over the sequence background (see Methods for definition), especially in the intermolecular context, that is between different protein chains in multi-chain WT simulations (*Figure 2—figure supplement 1A, B*). While glycine residues contribute significantly to the intermolecular interactions in WT with high absolute frequencies of G-Y and G-R contacts (*Figure 2B*), these contacts are either only slightly enriched over the expected sequence background (G-Y) or are even significantly depleted (G-R), as indicated in the corresponding source data. The latter suggests that in WT R and Y prefer to contact residues in the sequence other than G, which is also the case for single-chain interactions (*Figure 2—figure supplement 1B*).

Interestingly, R>K substitutions increase the importance of G-Y contacts with respect to multi-chain interactions, where they rank top and are even more enriched than K-Y contacts. At the same time, for both WT and R>K Lge1$_{1-80}$ variants the Y-Y contacts are enriched with respect to intramolecular interactions in single-chain simulations and become even more enriched in intermolecular interactions in multi-chain ones (*Figure 2—figure supplement 1A*). However, the opposite is seen for other homotypic contacts such as R-R in WT and Y>A or G-G in all three variants, since these contacts are enriched only for intramolecular interactions in single-chain simulations and are depleted for intermolecular interactions in multi-chain ones (*Figure 2—figure supplement 1B*). Altogether, our results show that R-Y is the top contact type in WT and together with Y-Y drives the multi-chain association of this IDR, in agreement with recent observations (*Bremer et al., 2022*). The results also point at a specific set of contacts which undergo significant rewiring when going from intramolecular interactions in single-chain simulations to intermolecular interactions in crowded systems, as further discussed below.

## Lge1$_{1-80}$ interaction valency is defined by its sequence composition

Interaction valency, defined as the average number of binding partners per protein molecule, stabilizes in the course of WT simulations and reaches the average value of approximately 4 over the last 0.3 µs (*Figure 2D*), with individual WT copies having anywhere between 1 and 9 partners at some point (*Figure 2E*). In contrast, multiple non-interacting protein copies are still observed for the two mutants throughout the simulations, with the average valencies plateauing around 2 and the maximum

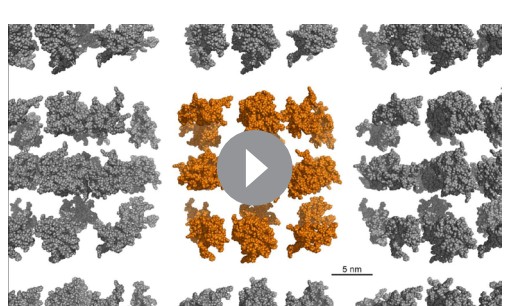

**Video 3.** Lge1$_{1-80}$ polypeptides self-associate in the crowded environment. All-atom, explicit-solvent molecular dynamics (MD) simulation with 24 copies, corresponding to the concentration of ~7 mM, of WT Lge1$_{1-80}$. The movie shows the complete 1 µs of the simulated trajectory with a 1 ns step. Proteins are shown in sphere representation. The 24 molecules are highlighted in orange, while the periodic images of the central simulation box are shown in gray. Note that the periodic images exhibit identical movements as the molecules in the central box. Scale bar corresponds to 5 nm.

https://elifesciences.org/articles/80038/figures#video3

number of partners not exceeding 6 in either case (*Figure 2E*). Thus, mutations with impaired phase separation as detected experimentally exhibit a significantly lower valency of protein-protein interactions in our simulations (see *Supplementary file 1* for details). Differences in valency can be also translated into different probabilities of contact formation, a key concept in percolation theory. We have estimated contact probabilities from simulations under the assumption of a well-mixed system, that is that all chains in the simulation box can in principle establish contacts with all the other chains. As shown in *Figure 2—figure supplement 1C*, contact probabilities evolve in direct proportion to valency, with a plateau over the last 0.3 µs. Notably, the WT contact probability reaches a level that is ~1.5-fold higher than for either mutant. This contact probability is sufficient to form a single percolating cluster with all 24 copies interconnected, and is thus higher than the critical contact probability at which one expects the network transition, which is not seen for the two Lge1$_{1-80}$ mutants (*Figure 2C*).

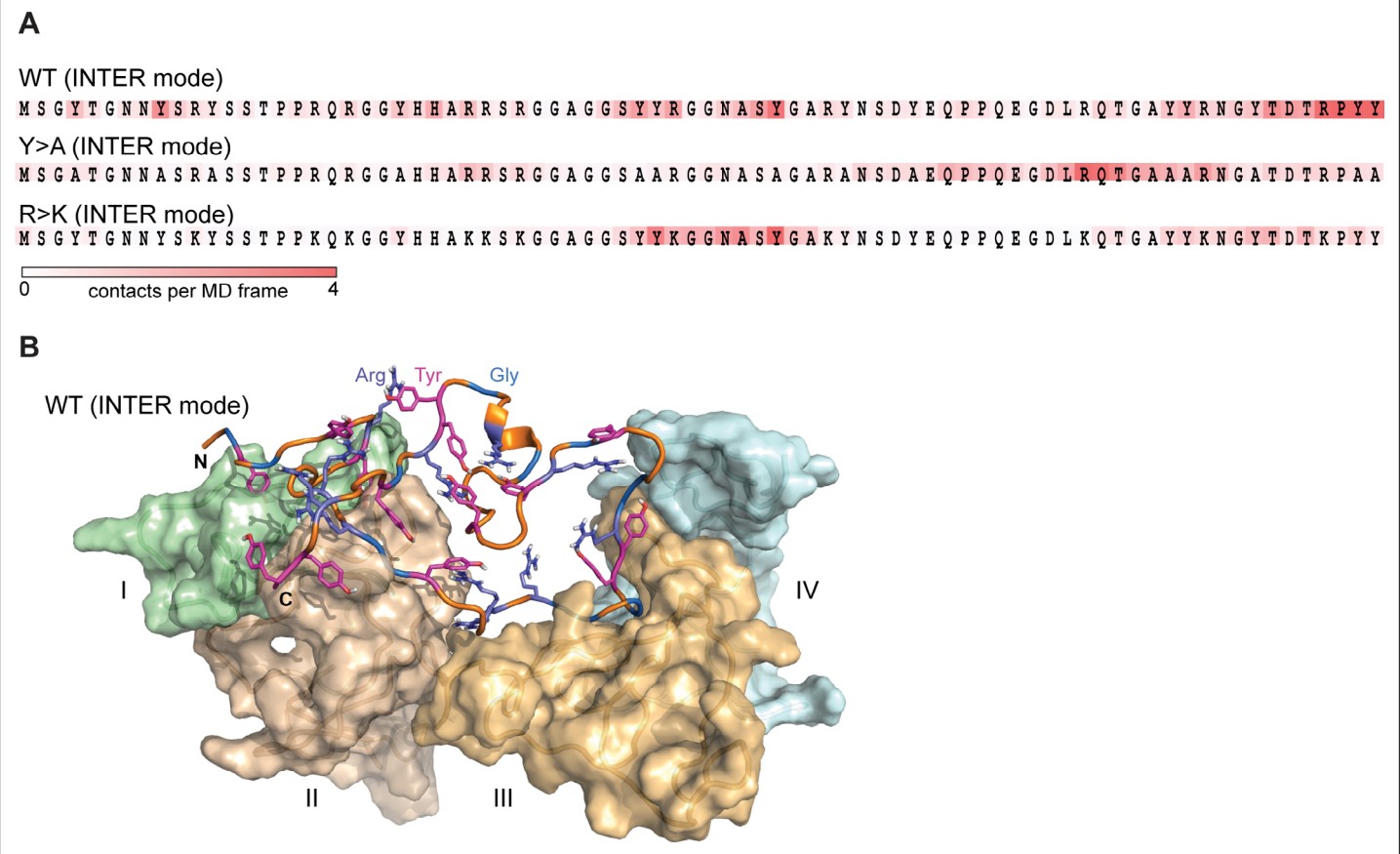

**Figure 3.** Lge1_{1-80} variants exhibit a dynamic binding mode in multi-chain systems. (**A**) Representative distributions of statistically defined interaction regions ('stickers') mapped onto the protein sequence. Protein sequences are colored according to the average contact statistics over the last 0.3 µs. Four interaction profiles of proteins having the number of partners corresponding to the average valency in the system (four partners for the Lge1_{1-80} WT and two partners for both mutants) and displaying the highest mutual correlations were used for the determination of the representative mode. The representative modes in all three cases display the Pearson correlation coefficient $R>0.6$ with the interaction profiles obtained by averaging over all 24 copies in each system. (**B**) 3D model of the representative binding mode for Lge1_{1-80} WT. An MD snapshot at 1 µs is given for a protein copy (shown in cartoon and sticks representation; the color scheme is the same as in *Figure 2B*) that is simultaneously interacting with four partners (shown in surface representation in pale cyan, pale green, wheat, and light orange, and indicated by Roman numerals).

The online version of this article includes the following source data and figure supplement(s) for figure 3:

**Source data 1.** Contact statistics used in panel A.

**Figure supplement 1.** Correspondence between inter- and intramolecular interaction modes of Lge1_{1-80} variants.

**Figure supplement 1—source data 1.** Contact statistics used in panel A and correlation data shown in panels B and C.

The interactions between IDPs in our simulations are characterized by a *dynamic binding mode* where the interacting sequence motifs ('stickers') and the non-interacting sequence motifs ('spacers') (*Martin et al., 2020*) can be described only statistically, and no well-defined, specific structural organization of protein complexes is detected (*Figure 3*). For WT, the typical binding mode, which corresponds to the average valency of 4, is characterized by different partners binding along the sequence in multiple, highly interactive regions (*Figure 3A and B*). In line with valency decrease, both mutants display fewer prominent interaction motifs, whereby for the phase separation-disruptive Y>A mutant intermolecular binding is detected for the C-terminal third of the molecule only (*Figure 3A*).

Interestingly, the dynamic modes identified in the intramolecular context are clearly different for all modeled proteins (*Figure 3—figure supplement 1A*), which is also partially observed at the level of pairwise contact statistics (*Figure 2—figure supplement 1A, B*), as discussed above. Specifically, intra- and intermolecular interactions rely on a similar pool of contacts by amino-acid type, but differ significantly if one analyzes specific sequence location of the interacting residues involved (*Figure 2—figure supplement 1A, B*). For example, one observes a high correlation between the frequencies of

different contacts by amino acid type when comparing intramolecular contacts in single-chain simulations and intermolecular contacts in multi-chain simulations (*Figure 3—figure supplement 1B*). This correlation is completely lost if one analyzes position-resolved statistics (2D pairwise contacts maps) or statistically defined interaction modes (*Figure 3A*, *Figure 3—figure supplement 1A, C*). Interestingly, the core of intramolecular interactions observed for a single molecule at infinite dilution and in the crowded context remain approximately the same, as reflected in the high correlation between intramolecular modes obtained in single- and multi-chain simulations (*Figure 3—figure supplement 1C*). The latter suggests that proteins maintain core self-contacts and establish new ones with neighbors, but do not lose self-identity as expected in a polymer melt. The observed 'symmetry breaking' between intra- and intermolecular mode of IDRs interactions is in line with the recent study by *Bremer et al., 2022*, and (*Martin et al., 2020*). In particular, Lge1$_{1-80}$ exhibits a relatively high net charge per residue (0.075), a non-uniform patterning of tyrosines ($\Omega_{aro}$ = 0.47, p = 0.57, see Martin et al. for methodological details), and a high abundance of arginines, all of which could contribute to symmetry breaking, as proposed in these studies.

## Lge1$_{1-80}$ sequence impacts its conformational behavior and dynamics

The perturbation of intra- and intermolecular interaction networks by mutations results in a different conformational behavior of the resulting Lge1$_{1-80}$ variants. At the level of single molecules, extensive interactions involving R and Y residues result in a substantial compaction of the WT chain with the avaerage radius of gyration $<Rg_{MD}>$ = 1.58 ± 0.12 nm (*Figure 4A*, see also *Supplementary file 1* for details). In contrast, the $Rg_{MD}$ distributions of the R>K and Y>A variants cover a wider range and display significantly higher average values (1.81 ± 0.35 and 1.71 ± 0.29 nm, respectively, *Figure 4A*). This is in all three cases more compact as compared with the predictions for a random coil of the same length ($<Rg_{rc}>$ = 2.50 nm, *Bernadó and Blackledge, 2009*). On the other hand, the difference between WT and the two mutants is directly related to the extent of intramolecular interactions: the looser the interaction network, that is the fewer long-range sequence contacts there are, the larger the $<Rg>$ (*Figure 4—figure supplement 1A*). In the crowded environment, WT again adopts a compact organization with an almost invariant $<Rg_{MD}>$ = 1.60 ± 0.22 nm when averaged over all 24 copies (*Figure 4B*), while the weakly self-interacting Y>A mutant unwinds toward more extended conformations ($<Rg_{MD}>$ = 1.97 ± 0.43 nm). At the same time, R>K, being most loosely packed in the single-molecule context, comes closer to WT values in the crowded environment ($<Rg_{MD}>$ = 1.67 ± 0.27 nm, *Figure 4B*). The latter can potentially be explained by the repulsive nature of K-K contacts, which are enriched in the single-molecule context and depleted in the crowded phase (*Figure 2—figure supplement 1A and B*). Note here that, unlike K-K pairs, R-R pairs can engage in π-π interactions as reflected in relatively higher fractions and enrichments of these contacts for Y>A, particularly in the single-molecule context (*Figure 2—figure supplement 1B*), as previously observed in PDB structures (*Vernon et al., 2018*). These results highlight the fact that the conformational behavior of IDPs in the bulk or in the crowded phase displays a clear sequence-specific character and cannot easily be generalized.

Single-molecule translational diffusion coefficients of Lge1$_{1-80}$ variants obtained from fitting of MSD curves with an applied finite-size PBC correction and solvent viscosity rescaling (see Methods for details) are ~120 μm$^2$/s for all three single-chain simulations or anywhere between 100 and 150 μm$^2$/s for multi-chain simulations and different Lge1$_{1-80}$ variants (*Figure 4C* and *Supplementary file 2*). In comparison, the diffusion constant of the similarly sized ubiquitin ($Rg$ = 1.32 nm, 76 aa, 8.6 kDa) at the protein concentration of 8.6 mg/ml is 149 μs$^2$/s (*Altieri et al., 1995*), while that of GFP ($Rg$ = 2.8 nm, 238 aa, 27 kDa) at the concentration of 0.5–3 mg/ml is ~90 μs$^2$/s (*Baum et al., 2014*). This suggests that the diffusional dynamics captured by our simulations may be realistic. The obtained viscosity values (see Methods for details) in single-chain simulations of the three Lge1$_{1-80}$ variants (effective concentration of 2.3 mg/ml) are all similar to each other and are close to the calculated solvent viscosity for TIP4P-D water/0.1 M NaCl of 0.83 mPa*s (*Figure 4—figure supplement 1F* and *Supplementary file 2*). In the crowded multi-chain systems (effective concentration of 6–7 mg/ml), the viscosity systematically increases by about 20% and is again similar for all three Lge1$_{1-80}$ variants (*Figure 4—figure supplement 1F* and *Supplementary file 2*). These calculated values are in the range of reported values for other similar systems, for example serum albumin (*Gonçalves et al., 2016*).

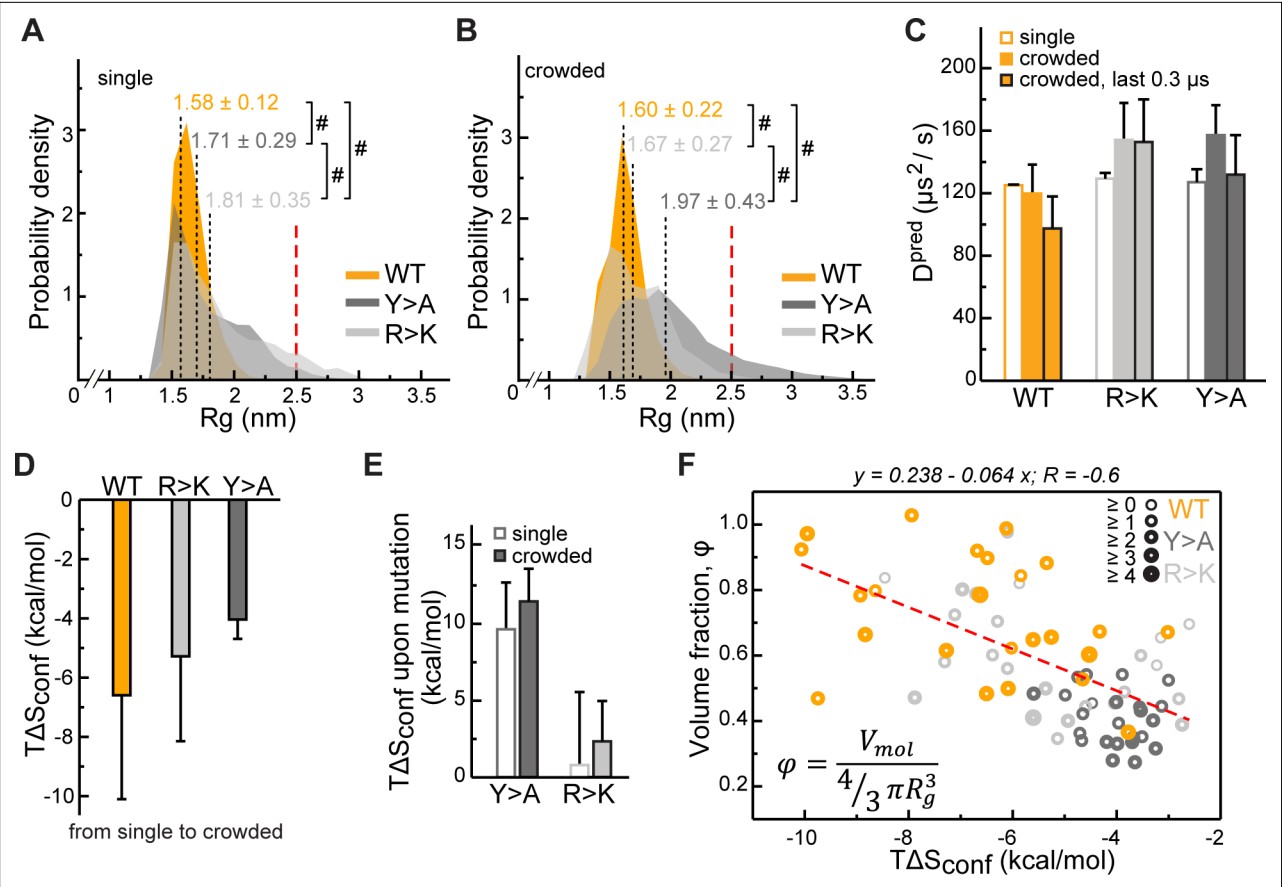

**Figure 4.** Impact of Lge1$_{1-80}$ sequence on its conformational behavior and dynamics. Distributions of radii of gyration (*Rg*) for Lge1$_{1-80}$ variants in (**A**) single-chain and (**B**) multi-chain systems. The last 0.3 μs of molecular dynamics (MD) trajectories were used to collect *Rg* statistics for (**A**) two independent runs of the single-chain simulation and (**B**) all 24 protein copies in the multi-chain system. Average *Rg* values from MD simulation (<*Rg*$_{MD}$>) and the corresponding standard deviations over the last 0.3 μs are indicated. Theoretical *Rg*$_{rc}$ value for an 80-aa disordered protein chain (see Methods) is shown with a vertical red dashed line. #, p-value <2.2 10$^{-16}$ according to Wilcoxon rank sum test with continuity correction. (**C**) MD-derived single-molecule translational diffusion coefficients of Lge1$_{1-80}$ variants. For single copies the values were averaged between the two independent MD runs. For 24 copy systems the value was averaged between all proteins. Error bars depict standard deviations. (**D**) Average changes in the configurational entropy (Δ*S*$_{conf}$) of a protein molecule for the transition from the single-molecule context (dilute state) to the crowded environment. Entropy values are given in energy units (*T*Δ*S*$_{conf}$, *T* = 310 K) and were obtained using complete 1 μs MD trajectories. Averaging was done for entropy differences in all possible combinations between two independent runs of single molecules and 24 protein copies in the crowded system. (**E**) Average changes in *T*Δ*S*$_{conf}$ upon different mutations (see Methods) in the single-molecule context and in the crowded environment. Averaging was done for entropy differences over all possible combinations (2×2 and 24×24, respectively). (**F**) Correlation between relative single-molecule configurational entropy changes (*T*Δ*S*$_{conf}$) and the corresponding average compactness (*φ*) values of all protein copies in the crowded system. The average valency of different protein copies is proportional to the thickness of the circles as given in the legend. Entropy values, average compactness, and average valency values were calculated over the complete 1 μs MD trajectories.

The online version of this article includes the following source data and figure supplement(s) for figure 4:

**Source data 1.** Molecular dynamics (MD) data used in panels A, B, D, E, and F.

**Figure supplement 1.** Conformational behavior and condensate rheology of Lge1$_{1-80}$ variants.

**Figure supplement 1—source data 1.** Molecular dynamics (MD) data used in panels B, C, D, and E.

In both single- and multi-chain simulations, the WT translational diffusion coefficients are somewhat lower than for either mutant (*Figure 4C* and *Supplementary file 2*). This effect does not appear to be related to protein size (<*Rg*$_{MD}$>, *Figure 4B*) or viscosity (*Figure 4—figure supplement 1F*), but may reflect protein slow-down due to more extensive interactions with partners, at least in the crowded environment (*Figure 2D*). For instance, the WT diffusion coefficient drops by approximately 20% over the last 0.3 μs of the trajectory (*Figure 4C*), which correlates with the formation of a single percolating cluster in the system (*Figure 2C*). At the same time, the R>K diffusion constant does not

change during the multi-chain simulation and is similar to the single-chain one (*Figure 4C*), likely due to electrostatic repulsion in the crowded environment. For Y>A there is no clear trend, whereby the diffusion constant over the last 0.3 μs of multi-chain simulations is similar to the single-chain one, which may be related to its larger size as compared to other variants (*Figure 4B*).

We have next quantified the effect of R and Y mutations on the Lge1$_{1-80}$ conformational dynamics by estimating the configurational entropy ($S_{conf}$) and its changes in different contexts via maximum information spanning tree (MIST) formalism. Due to the representation of molecules in internal bond-angle-torsion (BAT) coordinates (see Methods), MIST is well suited for unstructured protein ensembles, as shown before (*Fleck et al., 2018*). $S_{conf}$ displays a reasonable convergence between the individual replicas of the single-chain simulation on the 1 μs time scale (<0.1 kJ/mol/K), especially in the case of the weakly self-interacting Y>A mutant (*Figure 4—figure supplement 1B*). Interestingly, in the crowded multi-chain environment, we observe a significant decrease in $S_{conf}$ for all three variants, with the biggest change seen for WT and the smallest for Y>A (*Figure 4D*). These results suggest that the crowding involves a conformational reorganization of the molecules toward decreasing the available free volume, that is increasing their compactness (volume fraction) $\varphi$, defined here as the ratio between the van der Waals and the hydrodynamic volume of a molecule. Finally, there is a substantial increase in $S_{conf}$ associated with the Y>A mutation of the WT in both single- and multi-chain contexts, in contrast with the R>K mutation, where a much weaker effect is observed (*Figure 4E*).

In the crowded environment, $\varphi$ reaches plateau values over the last 0.3 μs (*Figure 4—figure supplement 1C*) with the average value fluctuating within a 2–4% interval for different averaging blocks (*Supplementary file 1*). Importantly, an increase in $\varphi$ in the crowded environment correlates directly with an unfavorable $\Delta S_{conf}$ (*Figure 4F*, Pearson $R$ is –0.6), an effect which can potentially be compensated for by the favorable enthalpy upon forming the extensive multivalent interaction network as observed for the WT. Conversely, $\Delta S_{conf}$ correlates poorly with the average number of bound partners (valency; Pearson $R$ is –0.3) and there is no correlation between the compactness $\varphi$, and the valency of interactions (not shown). Being mutually largely independent, these two characteristics of protein molecules – compactness and valency – are the key parameters describing the organization of the corresponding crowded phase, which generally reflect the entropic and the enthalpic contributions to self-organization, respectively.

## Describing condensate architecture via a fractal scaling model

To extrapolate the atomistic-level properties of the crowded protein phase to larger length scales, we modeled the assembly of phase-separated condensates as an iterative, fractal process (*Figure 5A*, see also Appendix 1). Colloid fractal models typically start with an Ansatz capturing the power-law dependence between mass and size across different scales (*Carpineti and Giglio, 1992*; *Lazzari et al., 2016*). For conceptual clarity and to demonstrate where the power-law dependence comes from, our derivation starts from a simple physical picture of associating clusters, and yields the known scaling relationship bottom-up. Thus, we assume that initially individual protein molecules, characterized by a given volume fraction $\varphi$, interact to form clusters of a given average valency $n$. In the next iteration, these clusters arrange into higher-order clusters, whereby the average valency of each cluster and the volume fraction occupied by the clusters formed in the previous iteration remain constant at all levels of organization. For instance, if a single protein molecule binds four other proteins ($n = 4$), this results in the second-iteration cluster consisting of 5 protein molecules (*Figure 5A*). In the next iteration, this 5-mer arranges with four other 5-mers into a new cluster with the same valency of 4, resulting in a larger cluster with 25 molecules. According to this model, the smaller clusters from iteration $i - 1$ always occupy the same volume fraction $\varphi$ of the apparent volume of the larger cluster from iteration $i$ (taken as 2/3 in *Figure 5A*). This scenario results in a simple fractal formalism whose benefit is that it yields exact solutions for different features of the clusters at each iteration (see Appendix 1). Thus, for each iteration $i$, the formalism returns the number of molecules (*Equation 1*), the apparent volume ($V_i$, *Equation 4*), the size ($R_i$, *Equation 5*), the mass ($M_i$, *Equation 8*), and the effective molar concentration of molecules in the cluster ($C_i$, *Equation 6*). While the above derivation for simplicity uses the constant volume fraction and valency at each iteration, we want to emphasize that in the case of structurally heterogeneous statistical fractal clusters, these parameters would be equivalent to their time- and ensemble-averaged counterparts.

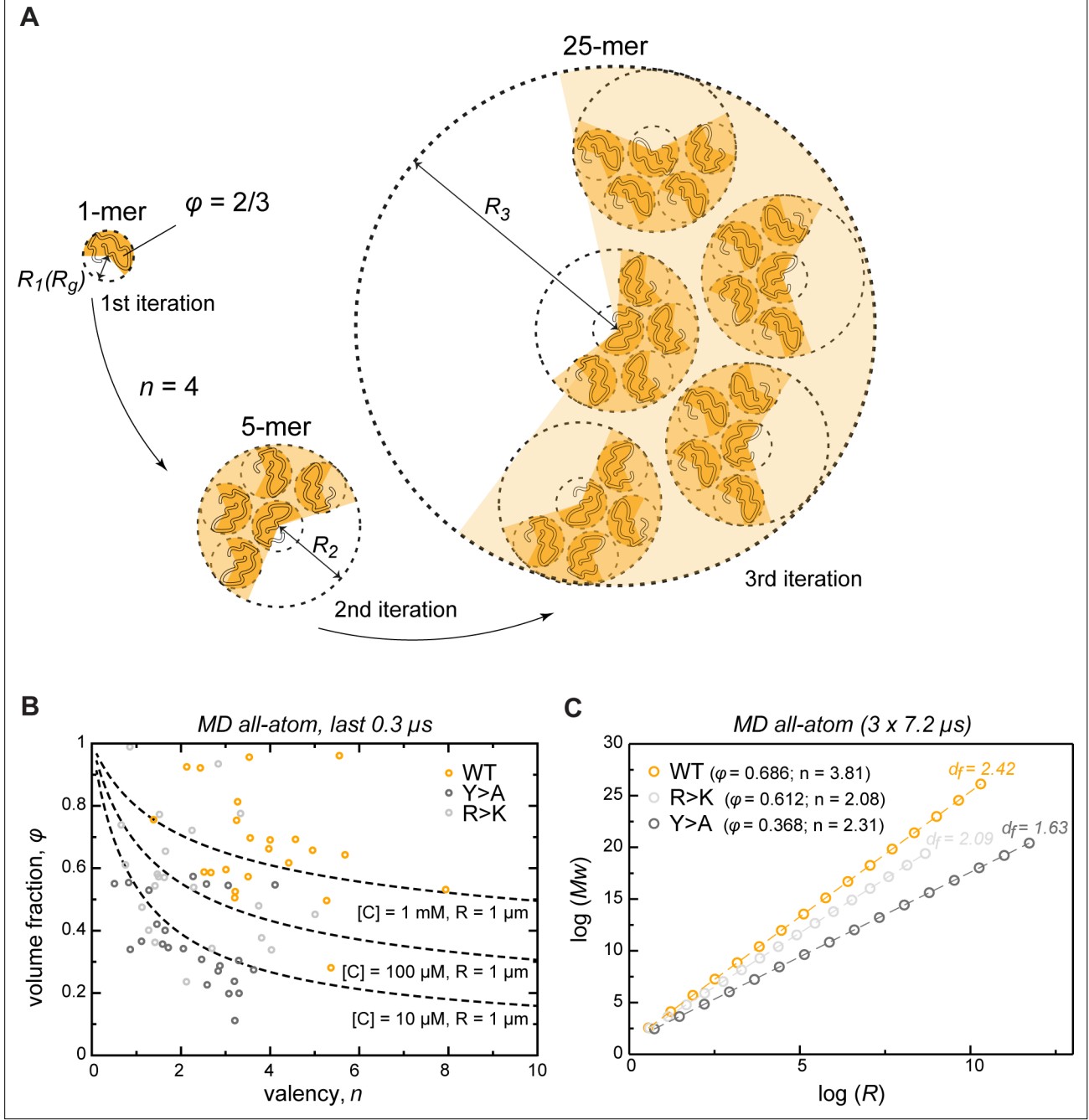

**Figure 5.** Describing condensate architecture via a fractal scaling model. (**A**) Schematic representation of a scaling principle in condensate assembly. Interaction valency ($n$) and compactness ($\varphi$) of individual proteins determine the properties of protein clusters at different iterations (see Appendix 1 for the corresponding formalism). (**B**) The parameter space for $\varphi$ and $n$ for a condensate with a fixed size ($R = 1$ µm) and concentration ([C]) described by the model (*dashed lines*). The $\varphi$ and $n$ for each individual protein molecule in the crowded environment averaged over the last 0.3 µs of molecular dynamics (MD) trajectories are shown with open circles. (**C**) Power-law dependence between mass and size of protein clusters at different iterations of the model with the applied valency and compactness corresponding to their average values over the 24 simulated protein copies and the last 0.3 µs of MD trajectories (indicated in the legend). Dashed lines show linear regression for the log $R$ vs. log $Mw$ plot with the corresponding slope or fractal dimension ($d_f$) indicated above the lines.

The online version of this article includes the following source data and figure supplement(s) for figure 5:

**Source data 1.** Numerical data used in panel B.

**Figure supplement 1.** Protein clusters and condensate topology of Lge1$_{1-80}$.

**Figure supplement 2.** Parameters used in the fractal model for different Lge1$_{1-80}$ variants (related to *Figures 5 and 6*).

**Figure supplement 2—source data 1.** Numerical data shown in the figure (fractal model parameters).

We have compared the predictions of the fractal model with what is seen in the actual Lge1$_{1-80}$ simulations. Importantly, the simulations in the first instance just give the average valency and compactness of individual chains in the dense phase. The fractal formalism, which is conceptually independent from the simulations, subsequently provides the dependence of condensate mass on its radius, $M(R)$, at any desired length scale. This, in turn, enables one to directly test the predictions of the fractal formalism in the case of the actual clusters seen in the simulations. Thus, over the last 0.3 μs of MD simulations, the WT multi-chain system displays a narrow distribution of sizes for the single molecule (*Figure 4B*) and leads to a single percolating cluster (*Figure 2D*, *Figure 5—figure supplement 1A*) with an average radius $R$ = 6.27 ± 0.24 nm. The latter point agrees closely with the predictions of the fractal formalism when the average values for $\varphi$ and $n$ obtained over the last 0.3 μs of MD simulations are used (*Figure 5—figure supplement 2*). Thus, a single cluster consisting of 24 chains observed in simulation directly corresponds to the third iteration of the model (*Figure 5—figure supplement 1A*). Moreover, the slope (*A*, *Equation 10*) and the intercept (*B*, *Equation 11*) of the linear regression for the log $R$ vs log $Mw$ plots are also similar between the simulations and the model (2.33 vs 2.42 and 1.13 vs 1.11, respectively) (*Figure 5—figure supplement 1B*). Finally, the latter parameters allow the exact calculation of the characteristic $\varphi$ and $n$ values (*Equations 12, 13*; 0.638 and 3.76, respectively), which again are very similar to those obtained directly from all-atom MD (*Figure 5—figure supplement 2*). These non-trivial correspondences suggest that fractal organization is present even at the shortest scale, that is at the level of MD simulation boxes.

The model also enables one to explore the space of $\varphi$ and $n$ parameters for a condensate with a fixed size and protein concentration (*Equation 7*, *Figure 5B*). For instance, for a 1 μm condensate with a 1 mM apparent protein concentration, the corresponding $\varphi$ and $n$ values are generally in the range observed in MD for the crowded WT system. Generally, in order to keep the ratio of size-to-concentration fixed, protein compactness must decrease non-linearly with increasing valency (*Figure 5B*, dashed lines). Accordingly, perturbation of compactness and valency due to the two types of studied mutations can result in a decrease of the apparent concentration in condensates of fixed size. Importantly, the compactness of IDPs is not a stable parameter and is tunable by different factors (temperature, pH, ionic strength, etc.) (*Uversky, 2009*). This, in turn, also suggests that IDP concentration inside condensates may also be adaptable and tunable. Thus, unwinding or compaction of an IDP due to any factor would change the apparent concentration and density in the condensates. The latter also opens up the possibility for potential microphase transitions inside of phase-separated droplets, in analogy to those known for lipid membranes (*Lewis and McElhaney, 2013*). Finally, protein concentration as a function of condensate size can be directly estimated from the model (*Figure 5—figure supplement 1C*). Notably, the complex topology of condensates (see also below) as proposed by the model allows for the formation of droplets with a very low apparent protein concentration.

## Valency and compactness define fractal dimension and condensate topology

The fractal model provides a direct relationship between size and mass of different clusters which captures the scaling behavior of the condensate matter. Thus, the slope of the line in the log $R$ vs. log $Mw$ plot (*A*) is equal to the fractal dimension $d_f$, which describes the topology of molecular clusters (*Carpineti and Giglio, 1992*). The fractal dimensions equaling exactly 1, 2, or 3 correspond to the objects exhibiting 1D, 2D, or 3D organization, respectively, while systems with non-integer $d_f$ have an intermediate dimensionality. The proposed model facilitates a direct investigation of the scaling properties in condensates for a molecule with defined characteristic compactness and valency of interactions. Most importantly, the fractal dimension of a condensate is completely defined by $\varphi$ and $n$ (*Equation 10*), reflecting the predictive potential of the proposed model. For instance, the average values of $\varphi$ and $n$ derived from MD simulations in the crowded state result in a different scaling behavior for WT, R>K and Y>A proteins, whereby their dimensionality respectively decreases (*Figure 5C*) and results in a different morphology of the corresponding condensates (*Figure 6B*), as observed experimentally (*Figure 1B and C*). Specifically, while the WT condensate simulated here exhibits a $d_f$ of 2.42 and is also experimentally found to undergo robust phase separation, the R>K mutant exhibits a lower $d_f$ of 2.09 and undergoes phase separation under a more limited set of conditions. Even more extremely, $d_f$ is 1.63 for the Y>A mutant (*Figure 5C*), which does not undergo phase separation and can be thought of as an object between 1D and 2D.

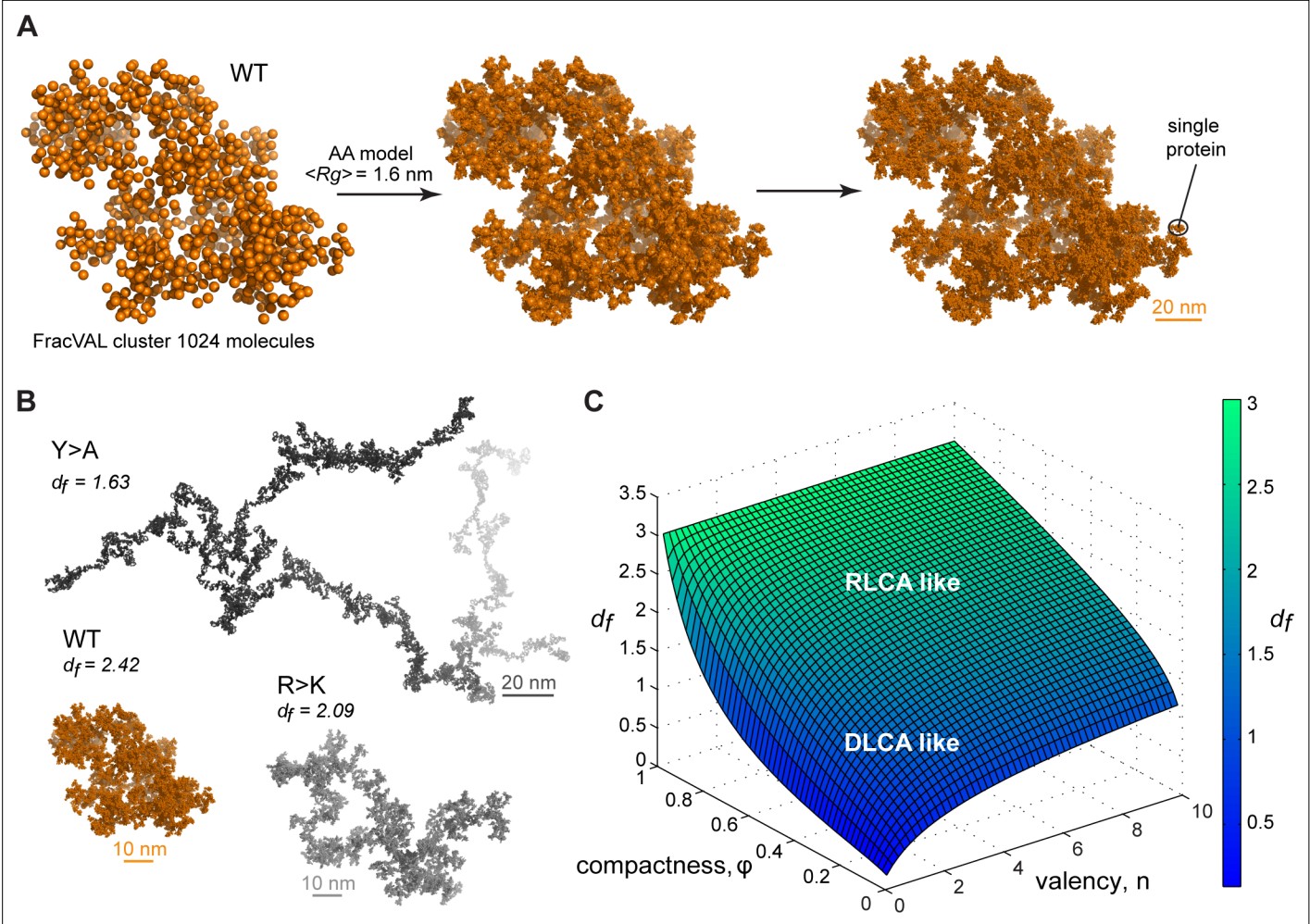

**Figure 6.** Reconstruction of the large-scale condensate architecture with atomistic resolution. (**A**) Transformation of a coarse-grained 1024 particle cluster obtained by FracVAL algorithm to an all-atom representation. The cluster was reconstructed using the fractal dimension $d_f$ and the averaged $Rg$ value derived from multi-chain simulations of Lge1$_{1-80}$ WT. (**B**) Representative 1024-protein clusters for Lge1$_{1-80}$ variants at all-atom resolution (see *Videos 4–6* to zoom in). (**C**) The non-linear dependence of the fractal dimension $d_f$ on $\varphi$ and $n$ as given by the model formalism (see *Equation 10*, Appendix 1). The surface is colored according to the corresponding $d_f$ values (see the scale bar).

The online version of this article includes the following source data and figure supplement(s) for figure 6:

**Source data 1.** Numerical data used in panel C.

**Figure supplement 1.** Partitioning of dextran of different sizes into condensates formed by Lge1$_{1-80}$ WT.

**Figure supplement 1—source data 1.** Dextran partitioning raw data used in panels A and B.

## Reconstructing the condensate architecture across scales

There exist several algorithms in the colloid literature that enable one to reconstruct the geometry of fractal clusters starting from a given fractal dimension $d_f$ (*Kätzel et al., 2008*; *Morán et al., 2019*; *Thouy and Jullien, 1994*) (and prefactor $k_f$, which is equal to 1 in the present model; see Appendix 1 for derivation). Recently, *Morán et al., 2019*, have proposed a robust and tunable algorithm, FracVAL, for modeling the formation of clusters consisting of polydisperse primary particles. We have used the values of $d_f$ derived from our simulations for the three Lge1$_{1-80}$ variants in combination with FracVAL to generate individual realizations of the respective condensate structure on the length scale of hundreds of nanometers. In *Figure 6A*, we demonstrate this procedure for WT Lge1$_{1-80}$: FracVAL produces cluster geometries using spherical particles with radii corresponding to the respective $<Rg_{MD}>$ values, which are then computationally replaced by the realistic protein conformations obtained from our simulations. In agreement with its micrometer-scale behavior observed in vitro, the modeled WT

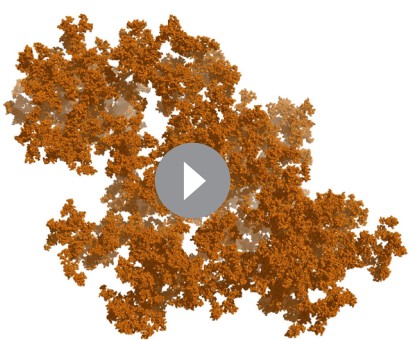

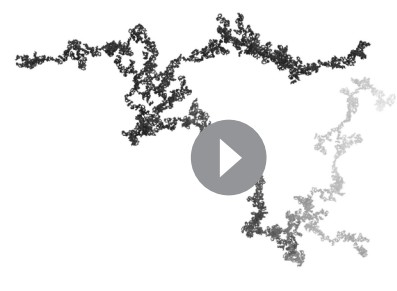

**Video 4.** Zoom-in of the internal organization of the WT Lge1$_{1-80}$ condensate at all-atom resolution. 1024-particle cluster was obtained by the FracVAL algorithm and transformed to an all-atom representation (see Methods). Protein atoms are shown as spheres. Video shows ×10 magnification.
https://elifesciences.org/articles/80038/figures#video4

**Video 6.** Zoom-in of the internal organization of the Y>A Lge1$_{1-80}$ condensate at all-atom resolution. 1024-particle cluster was obtained by the FracVAL algorithm and transformed to an all-atom representation (see Methods). Protein atoms are shown as spheres. Video shows ×30 magnification.
https://elifesciences.org/articles/80038/figures#video6

Lge1$_{1-80}$ condensate exhibits a densely packed fractal geometry. In contrast, the reconstructed Y>A cluster exhibits an elongated, filamentous topology of low dimensionality (*Figure 6B*), which may preclude the formation of well-defined phase-separated droplets (*Figure 1B* and *Figure 1—figure supplement 1B*). The R>K cluster exhibits an intermediate topology. The clusters shown in *Figure 6B* were generated using 1024 primary particles in all three cases: clearly, the Y>A cluster occupies a significantly larger volume as compared to the R>K cluster and especially the WT. In particular, the three reconstructed systems exhibit holes and cavities of different sizes (see *Videos 4–6* to zoom in the internal organization of the model condensates), with Y>A being most extreme in this regard. Finally, note that the three reconstructions shown in *Figure 6A and B* are individual snapshots of the local architecture; the full fractal model entails an ensemble of such snapshots, all configurationally different, yet still conforming to the same scaling pattern.

## Discussion

We have provided a multiscale description of Lge1$_{1-80}$ condensates extending from a detailed analysis of constituent molecules at the atomistic level all the way to the micrometer-sized droplets involving thousands of individual molecules, which can be observed in vitro. We have shown that mutations of R and Y residues induce perturbations at the level of both intra- and intermolecular interaction networks, and result in conformational effects that can be related to the phase behavior and the ability to form condensates as detected by light microscopy. Specifically, the characteristic descriptors of protein behavior in the crowded phase, valency, and compactness (volume fraction) are shown to be sufficient to describe the structural organization of condensates across length scales in the context of the proposed analytical fractal model. Importantly, the studied mutations substantially change either just the valency (R>K) or both the valency and compactness (Y>A) of Lge1$_{1-80}$ polypeptides as shown in MD simulations.

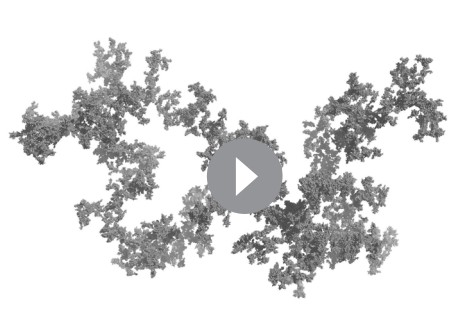

**Video 5.** Zoom-in of the internal organization of the R>K Lge1$_{1-80}$ condensate at all-atom resolution. 1024-particle cluster was obtained by the FracVAL algorithm and transformed to an all-atom representation (see Methods). Protein atoms are shown as spheres. Video shows ×15 magnification.
https://elifesciences.org/articles/80038/figures#video5

The applied simulation protocol reproduces the level of diffusive protein dynamics expected from molecules of Lge1$_{1-80}$ size. As a further indication of its general quality, the values of valency and compactness obtained from simulations are consistent with the difference in FRAP recovery dynamics observed for WT and R>K. Namely, the accurate fitting of FRAP data is possible only if using at least two components (*Figure 1—figure supplement 2*). According to *Sprague and McNally, 2005*, these components reflect the contribution of particle diffusion and interactions. Thus, the recovery in centrally bleached condensates is faster for WT than for the R>K mutant, which can be related to the higher compactness of WT particle across scales, as compared to R>K (*Figure 1E*, *Figure 1—figure supplement 2A*). On the other hand, the FRAP results for the condensates bleached in the peripheral area highlight the contribution of valency to condensate formation. Indeed, the recovery is about three times faster for the R>K mutant (*Figure 1—figure supplement 2B*), which could potentially be related to the lower valency of interactions and the ease of replacement of inactivated fluorescent species or/and exchange with proteins in the bulk. Indeed, a similar behavior with faster recovery for the R>K is observed when bleaching the whole condensate (*Figure 1E*, *Figure 1—figure supplement 2C*).

In the fractal model, the changes in protein valency and compactness translate to different scaling behavior and, subsequently, different topology of the condensates. Brangwynne and coworkers have successfully adopted the theoretical formalism of patchy colloids to capture the relative contributions of oligomerization, RNA binding, and structural disorder in the formation of stress granules and P-bodies (*Sanders et al., 2020*). In their analysis, they emphasized the role of the valency of colloid particles as a key parameter defining specificity and tunability of condensate features. Furthermore, Collepardo-Guevara and coworkers have used a coarse-grained patchy-particle colloid model to study the determinants of condensate stability and have highlighted the role of valency, while also clearly demonstrating the general impact of volume fraction (*Espinosa et al., 2020*). Our study now provides a general framework for assessing the importance of these two parameters on the formation of biomolecular condensates. Importantly, the proposed model predicts the existence and provides a quantitative characterization of topological properties of pre-percolation finite-size clusters that are in line with the recent findings (*Kar et al., 2022*; *Mittag and Pappu, 2022*; *Pappu et al., 2023*). More generally, the model provides the fractal dimension ($d_f$) of protein clusters and enables evaluation of different scale-dependent properties of clusters of arbitrary size, including protein density as a function of cluster size (*Figure 5—figure supplement 1C*, *Figure 5C*). Finally, MD simulations of proteins in the crowded context on the length scale of tens of nanometers can be used in combination with cluster-cluster aggregation algorithms to derive atomistically resolved models of the 3D organization of fractal clusters of any chosen size (*Figure 6A and B*, and *Videos 4–6*).

As discussed above, an emerging paradigm for biomolecular condensate formation is that of phase separation coupled to percolation. Importantly, fractal behavior, as explored in the present work, is naturally related to percolation phenomena. For instance, direct links between the fractal dimension defining scaling principles in self-similar clusters and critical exponents in different percolation models have been provided (*Kapitulnik et al., 1984*; *Stauffer et al., 1982*). While a similar formal derivation for our model is out of the scope of the present study, we can provide an illustration of how our results and the key parameters of the model can be interpreted according to percolation theory. A key concept in percolation theory is that of contact probability between components in the system. The network transition appears and a percolating cluster is formed if the contact probability exceeds a particular threshold, that is the critical contact probability ($p_{crit}$). For instance, in the simplest case according to the Flory-Stockmayer theory $p_{crit} = 1/(n-1)$, where $n$ is a number of bonds formed by each monomer, and is related to the valency in our model. Thus, interaction valency contributes directly to the spatial organization of pre-percolating clusters (the fractal dimension) and defines the threshold of the percolation. Therefore, our analysis shows that WT Lge1$_{1-80}$ displays a robust network transition due to high contact probability, which depends on its valency and, more generally, on the topology of its clusters. This perspective suggests that concentrations at which network transition is expected should be lower for WT than for either mutant, as indeed observed. Of note, the ability of IDRs to form low-dimensional fractal structures ($d_f$ is in a range of 1.6–1.9) upon the disruption of their tendency to phase separate by a polyalanine insertion was demonstrated experimentally for synthetic elastin-like polypeptides (*Roberts et al., 2018*).

The fractal formalism also provides a framework for approaching the question of protein concentration inside of phase-separated condensates, which covers a large range extending to tens of mM and beyond (*Brady et al., 2017*; *McCall et al., 2020*; *Ryan et al., 2018*). However, some proteins form condensates with extremely low concentrations in the dense phase. For example, the measured binodals of LAF-1 indicated that the concentration of the protein inside the droplet is 86.5 µM, which corresponds to an average separation between molecular centers of mass of 27 nm (*Wei et al., 2017*). Considering that the average $Rg$ of LAF-1 is approximately 4 nm, it is not immediately clear how the molecules inside the droplet are organized in order to simultaneously establish intermolecular contacts and also form low-density droplets. Fractal organization provides a simple resolution of this apparent conundrum. Namely, fractal systems are characterized by a remarkable property that their packing density is a function of the length scale on which it is examined (*Stanley, 1984*). For example, the density of a Sierpinski gasket, which is self-similar on all length scales, *decreases* exponentially with the length scale, with the exponent of $d_f$ - $d$, where $d$ is the dimensionality of the space. Translated to the question of biomolecular condensates, fractal organization enables high local concentration of biomolecules at short length scales and, simultaneously, low global concentration at long length scales, as illustrated in *Figure 5B* and *Figure 5—figure supplement 1C*.

Importantly, fractal systems are characterized by the existence of holes, that is unoccupied regions of space, whose size covers a wide range of length scales (*Stanley, 1984*). The observation that the µm-sized WT Lge1$_{1-80}$ droplets are fully permeable to dextrans in vitro (*Gallego et al., 2020*), even up to $Mw$ = 2000 kDa or $Rg$ ~27 nm (*Armstrong et al., 2004*; *Figure 6—figure supplement 1*), suggests that their organization allows for large holes (~55 nm in diameter or more). This is consistent with our fractal model, which proposes that the dimensionality of WT condensates is below 3 ($d_f$ = 2.42, *Figure 6*, *Video 2*). Finally, the multiscale nature of biomolecular condensates, as embodied in the statistical fractal model, also points to the possible formation of clusters with sizes well below the resolution limit of light microscopy. This may relate to some of the open questions in the condensate field, especially when it comes to their in vivo function (*McSwiggen et al., 2019*; *Musacchio, 2022*). Having said this, it should be emphasized that the proposed fractal model may be applicable to varying degrees in different systems. In particular, it was shown that in vitro Lge1 assembles a liquid-like core that is surrounded by an enzymatic outer shell formed by the E3 ubiquitin ligase Bre1 (*Gallego et al., 2020*). Whereas such condensates can exhibit a diameter of 1 µm or more when grown in vitro, they are expected to be smaller in cells (i.e. low nm range) (*Gallego et al., 2020*). Hence, the relevance of the proposed model will need to be tested experimentally with respect to nm-sized core-shell condensates in cells. Moreover, future studies must include the spherical Bre1 shell as a boundary condition which presumably constrains the 3D orientation of the Lge1 C-terminus and thereby impinges on the geometry of the Lge1 meshwork.

Finally, the fractal model predicts the coexistence of differently sized clusters within a condensate, as reported recently (*Kar et al., 2022*), which have a characteristic scaling of mass with condensate size in the nm to µm range. This prediction of the model can be tested using static light scattering (SLS) techniques and will be a subject of our future work: a linearly decreasing intensity as a function of the scattering vector in a log-log representation, as frequently seen for different colloidal systems, is expected by the fractal model (*Lazzari et al., 2016*; *Lin et al., 1989*). In fact, fractal dimension can be estimated from SLS experiments as the limiting value of scattering curves for high values of the product of the scattering vector $q$ and the average cluster size $<Rg>$ (*Hagiwara et al., 1996*). In addition, techniques such as DLS and MALS can be used in order to measure independently masses and sizes of LLPS condensates in vitro. It will be also important to analyze to what extent these features are retained in more complex, biologically relevant contexts.

Colloidal cluster formation is typically discussed in the context of two limiting regimes (*Klein et al., 1990*; *Lazzari et al., 2016*; *Lin et al., 1989*). In diffusion-limited cluster aggregation (DLCA), the rate of cluster formation is determined by the time it takes for colloidal particles to encounter each other and every encounter leads to binding. In reaction-limited cluster aggregation (RLCA), particles need to overcome a repulsive barrier before binding and not every encounter is productive. Both regimes result in fractal behavior, with DLCA leading to looser structural organization and lower fractal dimensions and RLCA leading to more compact structural organization and higher fractal dimensions. Computer simulations and scattering experiments show that $d_f$ is ~1.8 for DLCA and ~2.1 for RLCA (*Lazzari et al., 2016*; *Lin et al., 1989*). Importantly, Meakin and colleagues have shown that the two

regimes of colloid cluster formation are universal and do not depend on the chemical nature of the underlying particles, making them an attractive paradigm for modeling the multiscale structure of biomolecular condensates as applied here (*Lin et al., 1989*). However, for flexible, multivalent molecules like IDPs, the exact regime of cluster formation would depend on the values of $\varphi$ and $n$ and may be a tunable feature of the exact conditions. In *Figure 6C*, we present the non-linear relationship between the fractal dimension $d_f$ on $\varphi$ and $n$, which exhibits certain general trends. For example, high values of either compactness $\varphi$ or valency $n$ or both, such as in the case of WT Lge1$_{1-80}$, result in high values of $d_f$ and may be more associated with the RLCA model, while low values of both parameters may more be associated with the DLCA model (*Figure 6C*). A more quantitative analysis of the connection between the exact mechanism of cluster formation and the underlying parameters of compactness and valency will be a topic of future work.

Overall, our results provide an atomistic framework for understanding the role of valency and compactness of IDPs on condensate stability and architecture across scales. This presents an opportunity for the rational, quantitatively founded design of phase-separating agents with predefined condensate properties. Indeed, recent studies have demonstrated the possibility to tune condensate properties in vitro and in vivo by manipulating the aromatic residue content and molecular weight in IDPs (*Dzuricky et al., 2020*) or the size of disordered linkers and valency in modular proteins (*Lasker et al., 2021*). Finally, it is our hope that our results may help to critically embed the field of biomolecular condensates in the wider context of colloid chemistry. We expect that the powerful theoretical, computational, and experimental tools of colloid chemistry could propel the study of biomolecular condensates to the next level of fundamental understanding.

# Methods

## Key resources table

| Reagent type (species) or resource | Designation | Source or reference | Identifiers | Additional information |
|---|---|---|---|---|
| Gene (*Saccharomyces cerevisiae*) | LGE1 | SGD databank | YPL055C | Mutants used in this work were described in *Gallego et al., 2020* |
| Strain, strain background (*Escherichia coli*) | BL21 CodonPlus (DE3)-RIL | Stratagene | #200131 | Chemically competent cells |
| Other | (TRITC)-labeled dextran, Mw 155 KDa | Sigma-Aldrich | #T1287 | Final concentration 0.05 mg/ml |
| Other | (TRITC)-labeled dextran, Mw 65–85 KDa | Sigma-Aldrich | #T1162 | Final concentration 0.05 mg/ml |
| Other | (TRITC)-labeled dextran, Mw 2000 KDa | Thermo Fisher | #D7139 | Final concentration 0.05 mg/ml |
| Other | Dylight 488 NHS-Ester | Thermo Fisher | #46402 | Methods in this paper |
| Software, algorithm | ImageJ 1.53t | https://imagej.nih.gov/ij | | Version 1.53t |
| Software, algorithm | GraphPad Prism 7.0e | https://www.graphpad.com | | Version 7.0e |

## Protein expression and purification

All proteins were expressed in *Escherichia coli* BL21 CodonPlus (DE3) RIL cells. 6His-Lge1$_{1-80}$-StrepII constructs were induced by addition of 0.5 mM isopropyl 1-thio-β-D-galactopyranoside at OD$_{600}$ = 0.8 at 23 °C for 3 hr and purified as published elsewhere (*Gallego et al., 2020*) with Talon Superflow beads (Cytiva) in a final elution buffer (10 mM Tris, 1 M NaCl, 1 mM TCEP, 1 M imidazole, 10 % vol/vol glycerol, pH 7.5) and stored at –80 °C. For protein labeling with Dylight 488 NHS-Ester (Thermo Scientific), the final elution of the Lge1$_{1-80}$ constructs was performed in 10 mM HEPES, 1 M NaCl, 1 mM TCEP, 1 M imidazole, 10% vol/vol glycerol, pH 7.5. Labeling was performed during the elution step for 45 min according to the manufacturer's instructions. Unbound dye was removed by sequential buffer exchange in centrifugal filters Amicon Ultra 0.5 ml 3 K (Merk Millipore). Lge1$_{1-80}$-Dylight labeled protein was stored at –80 °C.

6His-LAF-1 was expressed and purified as described (*Elbaum-Garfinkle et al., 2015*) with some modifications as follows. Lysis buffer included 20 mM HEPES, 500 mM NaCl, 10% vol/vol glycerol,

14 mM β-mercaptoethanol, 10 mM imidazole, 1% vol/vol Triton 100, pH 7.5, and was supplemented with 0.5 mg/ml lysozyme, DNase I, and protein inhibitor mix HP (Serva). After washing and eluting from Ni-NTA Sepharose 6 FastFlow beads (GE Healthcare) in elution buffer (20 mM HEPES, 1 M NaCl, 10% vol/vol glycerol, 14 mM β-mercaptoethanol, 250 mM imidazole, pH 7.5), protein was labeled with Dylight 488 NHS-Ester (Thermo Scientific) for 30 min. Unbound dye was removed by sequential buffer exchange in centrifugal filters Amicon Ultra 0.5 ml 30 K (Merk Millipore). Finally, LAF-1-Dylight labeled protein was stored at –80 °C.

Protein quality was assessed by SDS-PAGE (4–12% gel, MOPS buffer) and Coomassie staining. Total purity of the protein was calculated by densitometry analysis of the gel bands with ImageJ. Percentage of the fraction of full-length protein was calculated in relation to all the bands present after purification.

## Solubility diagrams

Different concentrations of purified 6His-Lge1$_{1-80}$-StrepII proteins in a total volume of 100 µl were added to a bottom-clear 96-well plate (Greiner Bio-One) in a buffer containing 25 mM Tris, 100 mM NaCl, 100 mM imidazole, 1% vol/vol glycerol, and 1 mM DTT, pH 7.5. Varying concentrations of NaCl (100 mM to 3 M), imidazole (100 mM to 2.25 M), or tyrosine (0.25 mM to 2.5 mM) were analyzed as indicated in *Figure 1C*. Plates with the protein mix were incubated for 10 min at 20 °C. Turbidity of the samples was measured at 450 nm in a Victor Nivo plate reader (Perkin Elmer) at 20 °C. Assessment of protein phase separation or aggregation was performed by applying a total volume of 20 µl of the sample to a pretreated (*Gallego et al., 2020*) 16-well glass-bottom ChamberSLIP slide (Grace, BioLabs). DIC imaging was performed as previously described (*Gallego et al., 2020*).

## Circularity

Morphology of Dylight-labeled Lge1$_{1-80}$ particles was assessed by studying circularity, calculated using the formula:

$$Circ = 4\pi \frac{area}{perimeter^2}$$

Image analysis was done in Fiji/ImageJ by applying the particle analyzer-shape descriptor plugin, and statistical analysis was conducted in GraphPad Prism v 7.0e. For each construct, at least four independent images at each respective protein concentration were analyzed (1 µM for WT and 10 µM for R>K). Total number of the particles analyzed (*n*) is included in the figure legend (*Figure 1—figure supplement 1C*).

## Fluorescence recovery after photobleaching

FRAP experiments were performed on a temperature-controlled DeltaVision Elite microscope as previously described (*Gallego et al., 2020*). Lge1$_{1-80}$ WT-Dylight condensates were formed by 100% labeled protein at a final concentration of 10 µM in 25 mM Tris pH 7.5, 100 mM NaCl, 1% vol/vol glycerol, 1 mM TCEP, 100 mM imidazole. Lge1$_{1-80}$ R>K-Dylight was mixed with 50% of unlabeled protein (given the enrichment in lysine residues that are labeled) to a final protein concentration of 30 µM. LAF-1-Dylight was mixed with 30% of unlabeled protein and processed as described (*Elbaum-Garfinkle et al., 2015*) to a final concentration of 8 µM in 25 mM Tris, pH 7.5, 100 mM NaCl, 1 mM DTT. Bleaching was performed in protein condensates incubated for 30 min on pretreated 16-well glass-bottom slides (*Gallego et al., 2020*) by applying 20% power of a 50 mW laser 488 for 5 ms. Fluorescent intensity before bleaching was recorded for one frame prior to the bleach. Recovery of the bleach spot (central bleach, peripheral bleach, or whole condensate bleach) was recorded elapsed in time to avoid photobleaching (initially every 7 s, then 14 s, 30 s, and finally 60 s) with total 32 images for 20 min. Intensity traces were corrected for photobleaching. Recovery was calculated as published elsewhere (*Taylor et al., 2019*), normalized, and fitted to a double exponential function of the form:

$$C(t) = 1 - ae^{\frac{-t}{\tau(a)}} - be^{\frac{-t}{\tau(b)}}$$

Finally, the recovery half times were obtained by the numerical solution of the fitting equation:

$$C\left(t\right) = 1 - ae^{\frac{-t_{1/2}}{\tau\left(a\right)}} - be^{\frac{-t_{1/2}}{\tau\left(b\right)}} = 0.5$$

using WolframAlpha online (https://www.wolframalpha.com).

Total area of the bleached spot was calculated in ImageJ by relating it to the whole area of the condensate. Fluorescent intensity profiles for whole bleached condensates were acquired in ImageJ. For LAF-1, in addition to the double exponential function, several specific single exponent fits (*Taylor et al., 2019*) were tested in terms of quality of the fit.

## Dextran experiments

Experiments were performed as published elsewhere (*Gallego et al., 2020*). TRITC-Dextrans with a final concentration of 0.05 mg/ml were added to the samples containing 6His-Lge1$_{1-80}$-StrepII (2 µM) in a final buffer containing 25 mM Tris, 100 mM NaCl, 100 mM imidazole, 1% vol/vol glycerol, and 1 mM DTT, pH 7.5. Samples were incubated for 15 min at 20 °C on pretreated 16-well glass-bottom ChamberSLIP slides prior to imaging.

## Pairwise interaction free energy calculations

Pairwise interaction free energies were calculated using all-atom MC simulations according to the previously established framework (*Polyansky et al., 2009*). MC simulations were carried out in TIP4P water (*Jorgensen et al., 1983*), methanol, dimethylsulfoxide, and chloroform for the sidechain analogs of tyrosine, arginine, and lysine using OPLS force field (*Jorgensen et al., 1996*). Initial structures of the molecules were optimized in vacuo using AM1 semiempirical molecular orbital method (*Dewar et al., 1985*) and placed in rectangular boxes with explicit solvent. All calculations were performed with the BOSS 4.2 program (*Jorgensen and Tirado-Rives, 2005*) with periodic boundary conditions in the NPT ensemble at 298 K and 1 bar. Standard procedures were employed including Metropolis criterion and preferential sampling for the solutes (*Jorgensen and Ravimohan, 1985*). The potential of mean force computations for Y-Y, Y-K, and Y-R pairs were performed by gradually moving the solute molecules apart in steps of 0.05 Å along an axis defined by the particular atoms or centers of geometry, while both solute molecules were allowed to rotate around this axis. Non-bonded interactions were truncated with spherical cutoffs of 12 Å. The free energy changes for a particular pair were calculated in a series of consecutive MC simulations using statistical perturbation theory and double-wide sampling (*Jorgensen and Ravimohan, 1985*). Each simulation consisted of $3\times10^6$ configurations used for equilibration, followed by $6\times10^6$ configurations used for averaging.

## MD simulations

The full-length structure of Lge1 was modeled de novo using Phyre2 web portal (*Kelley et al., 2015*). The disordered N-terminal 1–80 aa fragment of Lge1 (Lge1$_{1-80}$) in the modeled structure lacked any secondary structure and was used as an initial configuration in further Lge1 simulations. All-atom MD simulations were performed for WT Lge1$_{1-80}$ fragment as well as for its Y>A and R>K mutants. Single-protein-copy MD simulations were carried out in $9\times9\times9$ nm$^3$ water boxes using two independent 1 µs replicas for each of the three Lge1$_{1-80}$ variants. The same initial configuration (see above) was used for all three. All systems had zero net charge and effective NaCl concentration of 0.1 M. Systems with 24 protein copies were simulated in $18\times18\times18$ nm$^3$ (WT, Y>A) or $19\times19\times19$ nm$^3$ (R>K) water boxes. Initial configurations for these simulations were generated as follows. Four different protein conformers were selected from the initial 100 ns parts of the two independent single-chain MD simulations (two conformations from each run) based on the criteria of having been the centers of the most highly populated clusters after clustering analysis performed using *cluster* utility (GROMACS) with the applied RMSD cutoff for backbone atoms of neighboring structures of 1.5 Å. The cells containing four copies were assembled manually and translated six times in different directions, resulting in a protein grid containing 24 protein copies. A total of 1 µs of MD statistics were collected for each large system. All MD simulations and the analysis were performed using GROMACS 5.1.4 package (*Abraham et al., 2015*) and Amber99SB-ILDN force field (*Lindorff-Larsen et al., 2010*). After initial energy minimization, all systems were solvated in an explicit aqueous solvent using TIP4P-D water model (*Piana et al., 2015*), which was optimized for simulating IDPs. The final NaCl concentration was 0.1 M (*Table 1*). The solvated systems were again energy-minimized and subjected to an MD equilibration of 30,000 steps

using a 0.5 fs time step with position restraints applied to all protein atoms (restraining force constants $Fx = Fy = Fz = 1000$ kJ/mol/nm) and 250,000 steps using a 1 fs time step without any restraints. Finally, production runs were carried out for all systems using a 2 fs time step. A twin-range (10/12 Å) spherical cutoff function was used to truncate van der Waals interactions. Electrostatic interactions were treated using the particle-mesh Ewald summation with a real space cutoff 12 and 1.2 Å grid with fourth-order spline interpolation. MD simulations were carried out using 3D periodic boundary conditions in the isothermal-isobaric (NPT) ensemble with an isotropic pressure of 1.013 bar and a constant temperature of 310 K. The pressure and temperature were controlled using Nose-Hoover thermostat (*Hoover, 1985*) and a Parrinello-Rahman barostat (*Parrinello and Rahman, 1981*) with 0.5 and 10 ps relaxation parameters, respectively, and a compressibility of $4.5 \times 10^{-5}$ bar$^{-1}$ for the barostat. Protein and solvent molecules were coupled to both thermostat and barostat separately. Bond lengths were constrained using LINCS (*Hess et al., 1997*).

Radii of gyrations (*Rg*) of simulated proteins were calculated using GROMACS *gyrate* utility, respectively. The average number of interaction partners per protein and the detailed statistics of intermolecular contacts were evaluated using GROMACS *mindist* and *pairdist* utilities with an applied distance cutoff of 3.5 Å, respectively, while intramolecular contact maps were generated using *mdmat* utility. Statistical significance of the difference between calculated parameters was evaluated using the Wilcoxon rank sum test with a continuity correction using R package (version 3.2.3). Protein structures were visualized using PyMol (*Schrodinger and DeLano, 2020*).

## Pairwise MD contact statistics and the dynamic interaction mode

Frequencies of pairwise contacts between different positions in protein sequences were collected over the last 0.3 μs of MD trajectories independently for every simulated protein chain. These frequencies can be represented as position-resolved 2D maps or can be collapsed as total interaction preferences at each position along the sequence (1D interaction profile or *dynamic interaction mode*). Finally, they can be grouped by contact type and converted to pairwise frequencies and enrichments. An enrichment for a pairwise contact *A-B* is calculated as:

$$ENR\left(A, B\right) = \frac{f\left(A, B\right)_{MD}}{f\left(A, B\right)_{exp}},$$

where $f_{MD}$ is an observed MD frequency of contacts between *A* and *B* and $f_{exp}$ is the frequency of such contacts expected at random, given the sequence composition of the chain, that is $f\left(A, B\right)_{exp} = f\left(A\right) \times f\left(B\right)$, where $f\left(A\right)$ is the frequency of *X* in the sequence. Individual 1D interaction profiles were obtained for each simulated protein considering only intramolecular protein contacts (INTRA) or only contacts with partners (INTER). Four individual interaction profiles of proteins having the number of partners corresponding to the average valency in the system over the last 0.3 μs and displaying the highest mutual correlations were used for the determination of the representative INTER mode. Interaction profiles averaged between the individual MD replicas of single-chain simulations were used for the determination of a representative INTRA mode.

## Theoretical estimate of *Rg*

The estimation was done according to the scaling model used in the polymer theory to connect *Rg* and the chain length of a random coil as follows:

$$Rg = R_0 N^{\upsilon},$$

where *N* is the length of Lge1$_{1-80}$ ($N = 80$) and $R_0$ and $\upsilon$ are the empirical parameters refined for IDPs ($R_0 = 0.254$ nm; $\upsilon = 0.522$) (*Bernadó and Blackledge, 2009*).

## Cluster analysis

The largest protein-protein interaction clusters in the 24-copy simulated system were identified using hierarchical clustering. For this purpose, minimum-distance matrices were calculated from each MD trajectory sampled at every 100 ps using GROMACS *mindist*. The clustering was done in MATLAB (R2009) using function *cluster* with an applied distance cutoff of 3.5 Å.

## Entropy calculations

The configurational entropy was evaluated by applying the MIST approximation (*King et al., 2012*) using the PARENT suite (*Fleck et al., 2016*), a collection of programs for the computation-intensive estimation of configurational entropy by information theoretical approaches on parallel architectures. All MD trajectories were first converted from Cartesian to BAT coordinates. To assess the convergence of configurational entropy ($S_{conf}$), cumulative plots were generated for single copy systems using a 50 ns time step. Due to the relatively slow convergence of the entropy values (*Figure 4—figure supplement 1B*), the final entropy calculations were performed for the entire 1 µs trajectories. Note that the absolute $S_{conf}$ values are negative and carry arbitrary units due to the exclusion from the calculations of the constant momentum part of the configurational entropy integrals and are reported just to illustrate the convergence of the entropy values as a function of simulated time. However, upon subtraction of these absolute values (i.e. for single and crowded systems), the relative entropy ($\Delta S_{conf}$) carries correct physical units and is equal to the total configurational entropy change between the two systems. The relative entropies of a protein in single-chain and multi-chain systems were averaged over all possible combinations of the two single-protein copies and the 24 crowded-system copies. To estimate the effect of mutations on configurational heterogeneity, the corresponding entropy differences were calculated as follows:

$$\Delta S_{conf}^{mut} = \left[ \frac{S_{conf}^{mut}}{(3N_{mut} - 6)} - \frac{S_{conf}^{WT}}{(3N_{WT} - 6)} \right] (3N_{mut} - 6)$$

where $N_{mut}$ and $N_{WT}$ represent the numbers of atoms in mutant and WT proteins, respectively. The final values of configurational entropy differences were multiplied by the temperature (*T* = 310 K) and converted to kcal/mol units.

## Estimation of diffusion coefficients and shear viscosity

Diffusion coefficients of individual protein chains were calculated following the procedure described elsewhere (*von Bülow et al., 2019*), together with viscosity estimation (*Hess, 2002*), application of corrections for size-dependent effects (*Yeh and Hummer, 2004*), and rescaling against experimentally comparable values (*Fennell et al., 2018*).

Thus, translational diffusion coefficients $D_t^{PBC}$ were extracted for individual molecules by analyzing center-of-mass mean-square displacement (MSD) curves, considering that:

$$MSD = c + 6D_t^{PBC}\tau$$

for $\tau$ approaching infinity. The above equation was fitted in a linear regime of MSD between 20 ns and 40 ns for the 24-copy systems (*Figure 4—figure supplement 1D*), and between 5 ns and 15 ns for the single molecule. As previously suggested (*Yeh and Hummer, 2004*), the thus obtained diffusion coefficients were corrected for size-dependent effects that arise from periodic boundary conditions (PBC). Applying this correction, the diffusion coefficient $D_t$ can be determined as:

$$D_t = D_t^{PBC} + \frac{k_B T \xi}{6\pi \eta L},$$

where *L* is the edge length of the simulation box, $\eta$ is the viscosity of the system that the particle is simulated in, and $\xi$ = 2.837297, a term arising from the cubic lattice (*Yeh and Hummer, 2004*). The latter correction requires estimation of shear viscosity values in the system. For this purpose, short 10 ns NVT MD simulation was performed for each system starting from the last snapshot of 1 µs simulations with detailed output for GROMACS energy file (every 10 fs). Shear viscosities were extracted from these NVT simulations using the Green-Kubo formula (*Hess, 2002*):

$$\eta_{ij} = \frac{V}{k_B T} \int_0^\infty C_{ij}(t) \, dt$$

where *V* denotes the volume of the simulation box and $C_{ij}(t)$ is the autocorrelation function:

$$C_{ij}(t) = \langle P_{ij}(t) P_{ij}(0) \rangle$$

of the pressure tensor elements $P_{ij} = P_{xy}, P_{xz}, P_{yz}, \frac{P_{xx}-P_{yy}}{2}, \frac{P_{xx}-P_{zz}}{2}, and \frac{P_{yy}-P_{zz}}{2}$ .

The autocorrelation function was numerically integrated between 0 ps and 1 ps, followed by analytical integration up to infinity. The analytical part of the integral was determined by a double exponential fit of the data between 1 ps and 5 ps (*Figure 4—figure supplement 1E*):

$$C_{ij}(t) = ae^{\frac{-t}{b}} + ce^{\frac{-t}{d}}$$

Shear viscosity $\eta$ was then determined by averaging over the $\eta_{ij}$ of the evaluated pressure tensor elements. Finally, the corrected diffusion coefficients were rescaled by the ratio of the simulated and experimentally determined water viscosities (*Fennell et al., 2018*):

$$D^{pred} = D_t \cdot \frac{\eta_{sim}}{\eta_{expt}}$$

For the experimental water viscosity value at 310 K and 0.1 M salt $\eta_{expt}$ of 0.69 mPa·s (*Fennell et al., 2018*) was used. The simulated value of viscosity ($\eta_{sim}$ = 0.83 mPa·s) in TIP4P-D water box at the same conditions was obtained previously by us (in preparation) using a series of 100 ns NVT MD simulations for cubic boxes of different size (3, 4, and 5 nm).

MSD curves for complete 1 µs MD trajectories or only their last 0.3 µs fragments were calculated using *msd* utility from the GROMACS package. Pressure tensors were obtained using *energy* utility from the GROMACS package for the analysis of NVT simulations. Viscosities were calculated as described above using MATLAB (R2009) scripts written specifically for this purpose.

## Modeling and visualization of condensates topology

To generate a visual representation of a condensate with a fractal dimension obtained by the self-propagation model (see Appendix 1), FracVAL algorithm was used (*Morán et al., 2019*). FracVAL is a tunable algorithm for generation of fractal structures of aggregates of polydisperse primary particles, which preserves the predefined fractal dimension ($d_f$) and the fractal prefactor ($k_f$) to generate aggregates of desired size. The scaling law in this case can be defined as shown in *Equation 14*. The prefactor $k_f$ is equal to 1 in the present model (see Appendix 1). The $d_f$ values for generating cluster models in FracVAL were calculated according to *Equation 10* using the averaged $\varphi$ and $n$ values over the last 0.3 µs of the 24-copy MD simulations, while <$Rg_{MD}$> values over the last 0.3 µs were taken as effective sizes of primary particles (detailed parameters are listed in *Figure 5—figure supplement 2*). For visualization purposes, the size of condensates generated by FracVAL was defined as 1024 molecules. FracVAL cluster models were transformed to all-atom resolution by using selected protein MD conformations with the representative $Rg$ values corresponding to the respective <$Rg_{MD}$> and scripts specially written for this purpose. The obtained coarse-grained and all-atom structures were visualized using PyMol (*Schrodinger and DeLano, 2020*).

## Acknowledgements

This work was supported by Austrian Science Fund FWF Standalone Grants P 30550 and P 30680-B21 (to BZ); a NOMIS Pioneering Research Grant and a grant of the Austrian Science Fund (FWF, project F79) (to AK).

## Additional information

### Funding

| Funder | Grant reference number | Author |
| --- | --- | --- |
| Austrian Science Fund | P 30550 | Bojan Zagrovic |
| Austrian Science Fund | P 30680-B21 | Bojan Zagrovic |
| NOMIS Stiftung | Pioneering Research Grant | Alwin Köhler |

| Funder | Grant reference number | Author |
| --- | --- | --- |
| Austrian Science Fund | F79 | Alwin Köhler |
| Ministry of Science and Higher Education of the Russian Federation | agreement No. 075-15-2020-773 | Roman G Efremov |

The funders had no role in study design, data collection and interpretation, or the decision to submit the work for publication.

### Author contributions

Anton A Polyansky, Conceptualization, Formal analysis, Investigation, Visualization, Methodology, Writing – original draft, Writing – review and editing; Laura D Gallego, Conceptualization, Formal analysis, Investigation, Visualization, Writing – original draft, Writing – review and editing; Roman G Efremov, Investigation; Alwin Köhler, Resources, Supervision, Writing – review and editing; Bojan Zagrovic, Conceptualization, Supervision, Funding acquisition, Writing – review and editing

### Author ORCIDs

Anton A Polyansky (iD) https://orcid.org/0000-0002-1011-2706

### Decision letter and Author response

Decision letter https://doi.org/10.7554/eLife.80038.sa1
Author response https://doi.org/10.7554/eLife.80038.sa2

## Additional files

### Supplementary files

- Supplementary file 1. Technical summary on molecular dynamics (MD) results.
- Supplementary file 2. Diffusion summary.
- MDAR checklist

### Data availability

All data generated or analysed during this study are included in the manuscript and supporting files (Supplementary Files 1 and 2); source data files have been provided for Figure 2 (Figure 2 —source data 1), Figure 1—figure supplement 1 (Figure 1—figure supplement 1—source data 2), Figure 1—figure supplement 2 (Figure 1—figure supplement 2—source data 1), Figure 5—figure supplement 2 (Figure 5—figure supplement 2—source data 1); compressed folders containing source data files have been provided for Figure 1 (Figure 1 —source data 1), Figure 2 (Figure 2 —source data 2), Figure 3 (Figure 3 —source data 1), Figure 4 (Figure 4 —source data 1), Figure 5 (Figure 5 —source data 1), Figure 6 (Figure 6 —source data 1), Figure 1—figure supplement 1 (Figure 1—figure supplement 1—source data 1), Figure 2—figure supplement 1 (Figure 2—figure supplement 1—source data 1), Figure 3—figure supplement 1 (Figure 3—figure supplement 1—source data 1), Figure 4—figure supplement 1 (Figure 4—figure supplement 1—source data 1), Figure 6—figure supplement 1 (Figure 6—figure supplement 1—source data 1). These source files contain the numerical data used to generate the figures.

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

## Appendix 1

### Fractal model of condensate assembly

The number of molecules in a cluster at iteration $i$ can be calculated according to a simple geometrical progression:

$$(n + 1)^{i-1} \tag{1}$$

where $n$ is the valency of interactions or more generally – the coordination number. An apparent volume $v$ of a single molecule, whereby atoms occupy the fraction $\varphi$ of that volume, can be expressed as:

$$v = \frac{V_{mol}}{\varphi} \tag{2}$$

where $V_{mol}$ is the molecular volume, which in turn is proportional to the molecular weight ($M_W$) with a factor $\kappa$ ($\kappa = 1.21$ used for all calculations *Harpaz et al., 1994*):

$$V_{mol} = \kappa M_W \tag{3}$$

Thus, combination of *Equations 1 and 2* allows to estimate an apparent volume of condensate ($V_i$) at iteration $i$:

$$V_i = \frac{(n + 1)^{i-1}}{\varphi^i} V_{mol} \tag{4}$$

Under the assumption of a spherical geometry, a characteristic size of the condensate at iteration $i$ ($R_i$) can be derived from *Equation 4*:

$$R_i = \sqrt[3]{\frac{3}{4} V_{mol} \frac{(n + 1)^{i-1}}{\pi \varphi^i}} \tag{5}$$

An effective molar concentration ($C_i$) of molecules in a condensate at iteration $i$ then can be estimated as:

$$C_i = \frac{\varphi^i}{V_{mol} N_a} 10^{-3} m^3 \tag{6}$$

where, $N_a$ is the Avogadro number. A combination of *Equations 5 and 6* gives the volume fraction as a function of valency for a condensate with a given concentration ($C$) and size ($R$):

$$\varphi(n) = e^{\dfrac{ln\left(V_{mol}N_a 10^3 m^{-3}\right)}{1 + \dfrac{ln\left(\frac{4}{3}\pi R^3 C N_a 10^3 m^{-3}\right)}{ln(n + 1)}}} \tag{7}$$

The above formalism allows one to extract the characteristic parameters $\varphi$ and $n$ from the linear regression of an empirical log $R$ vs. log $M$ plot. First, the mass $M_i$ of a condensate at iteration $i$ is given as:

$$M_i = M_W (n + 1)^{i-1} \tag{8}$$

A combination of *Equations 5 and 8* then gives analytical expressions for a slope ($A$) and an intercept ($B$) for the linear regression plot:

$$logM = AlogR + B \tag{9}$$

$$A = \frac{3}{1 - \frac{log\varphi}{log\,(n+1)}} \tag{10}$$

$$B = \frac{log\frac{4\pi}{3\kappa} - logM_W\frac{log\varphi}{log\,(n+1)} + log\varphi}{1 - \frac{log\varphi}{log\,(n+1)}} \tag{11}$$

Finally, a combination of **Equations 10 and 11** gives equations for $\varphi$ and $n$ using the slope $A$ and the intercept $B$ for the linear regression (9):

$$\varphi = e^{\frac{3B + (A-3)\,logM_W}{A} - log\frac{4\pi}{3\kappa}} \tag{12}$$

$$n = \varphi^{\frac{A}{A-3}} - 1 \tag{13}$$

Note that $A$, also known as the 'fractal dimension' ($d_f$) (**Carpineti and Giglio, 1992**), is the key parameter describing the condensate organization. The scaling law in this case is typically defined as:

$$N = k_f \left(\frac{R}{R_g}\right)^{d_f} \tag{14}$$

where $N$ is the number of molecules in each cluster with the corresponding size $R$, while $R_g$ is an effective size of an individual molecule, and $k_f$ is the fractal prefactor. In the present model of condensate assembly, $k_f$ is equal to 1, as proven below. Thus, according to **Equation 14** the number of molecules for a cluster obtained at iteration $i$ can be calculated as:

$$N_i = k_f \left(\frac{R_i}{R_1}\right)^{d_f}$$

where $d_f$ is a slope ($A$) for log $R$ vs. log $M$ linear regression and is defined in **Equation 10**. A combination of **Equations 1 and 5** gives:

$$N_i = (n+1)^{i-1}$$

$$R_i = \sqrt[3]{\frac{3}{4}V_{mol}\frac{(n+1)^{i-1}}{\pi\varphi^i}}$$

$$R_1 = \sqrt[3]{\frac{3}{4}\frac{V_{mol}}{\pi\varphi}}$$

$$\frac{R_i}{R_1} = \sqrt[3]{\frac{(n+1)^{i-1}}{\varphi^{i-1}}} = \left(\frac{(n+1)^{i-1}}{\varphi^{i-1}}\right)^{\frac{1}{3}}$$

then simplification leads to:

$$(n+1)^{i-1} = k_f \left(\frac{(n+1)^{i-1}}{\varphi^{i-1}}\right)^{\frac{A}{3}}$$

$$k_f = \frac{\left(\varphi^{i-1}\right)^{\frac{A}{3}}}{\left((n+1)^{i-1}\right)^{\frac{A}{3}-1}}$$

and by using *Equation 13* to substitute $n+1$ one derives:

$$k_f = \frac{\left(\varphi^{i-1}\right)^{\frac{A}{3}}}{\left(\left(\varphi^{\frac{A}{A-3}}\right)^{i-1}\right)^{\frac{A}{3}-1}}$$

$$k_f = \frac{\left(\varphi^{i-1}\right)^{\frac{A}{3}}}{\left(\varphi^{i-1}\right)^{\frac{A-3}{3}\frac{A}{A-3}}}$$

$$k_f = \frac{\left(\varphi^{i-1}\right)^{\frac{A}{3}}}{\left(\varphi^{i-1}\right)^{\frac{A}{3}}}$$

$$k_f \equiv 1$$

