## [Editor Report]

In this work, the authors introduce and develop upon a computational model to investigate and quantify the effect of protein conformations and valence of interaction sites as organizers of structure within biomolecular condensates. The authors integrate their findings with new and emerging concepts regarding the coupling between phase separation and percolation as a determinant of driving forces and internal organization of condensates. The key insight that emerges from the current work pertains to the structure that prevails across length scales.

---

## [Decision Letter]

**Decision letter after peer review:**

Thank you for submitting your article "Protein compactness and interaction valency define the architecture of a biomolecular condensate across scales" for consideration by *eLife*. Your article has been reviewed by 2 peer reviewers, and the evaluation has been overseen by a Rohit Pappu as Reviewing Editor and José Faraldo-Gómez as Senior Editor. The reviewers have opted to remain anonymous.

The Reviewing Editor has drafted this to help you prepare a revised submission.

Essential revisions:

1) Many of the experimental details need careful scrutiny. Both reviewers raise specific concerns and make specific requests. Please respond with all the details that the reviewers are requesting.

2) Both reviewers are concerned about the rather sweeping generalizations and statements made that do not square with the state of the art. First and foremost, it is now clear that the phase transitions in question are not purely segregative LLPS type phenomena. That the field has overused this term and done so without a care is not a good enough reason to perpetuate the false notion of a pure LLPS type behavior. One of the earlier studies demonstrating the coupling of segregative transitions viz., phase separation, and associative transitions, viz., percolation, was published in *eLife* and elsewhere. Please see: https://elifesciences.org/articles/30294 and http://iopscience.iop.org/article/10.1088/1367-2630/aab8d9. Given the considerable progress made by several labs, and especially those of Mittag and Pappu on the topic of IDR phase transitions, it is imperative that the motivation for the current work not be that LLPS is "poorly understood" (see comments by Reviewer 2).

3) Please provide a coherent motivation/justification for co-opting concepts such as the fractal analysis from colloidal chemistry for the analysis of the simulations. Please note that this is not the first time fractal analyses have been brought to bear in studying phase separating IDRs. They've been previously deployed in studies of ultra-coarse grained simulations of mimics of the exon 1 encoded region of huntingtin (http://www.sciencedirect.com/science/article/pii/S0006349514007371).

4) The issues of convergence, statistical robustness, and finite size effects need careful consideration. How do the images interact with one another in the dilute and dense phase simulations. For simple liquids, even the earliest simulations deployed ca. 100 molecules for querying properties of neat liquids. How then does one justify the use of 24 copies for a complex fluid? In lattice-based simulations, the effects of finite size were systematically analyzed, the inference was that one needs at least 100+ molecules to get to coherent descriptions of two coexisting phases. The current work does not simulate coexisting phases but approaches each phase separately. So, fewer molecules are reasonable, but it must still be the case that a reasonably rigorous assessment of finite size effects is provided.

5) Regarding the differences between Arg and Lys, the sources of differences are intrinsic and context dependent. As Reviewer 2 notes, this is not as enigmatic as the authors note. Please see recent contributions that have demonstrated clear differences of Arg vs. Lys as drivers of speckle formation (https://doi.org/10.1016/j.molcel.2020.01.025), the realization that Arg and Lys are very different in terms of their intrinsic free energies of hydration (https://pubs.acs.org/doi/abs/10.1021/acs.jpcb.1c01073), and that these differences contribute directly to the cation-specificity of IDP conformational ensembles (https://www.pnas.org/doi/full/10.1073/pnas.2200559119). These physical principles and the distinction of Arg being sticky vs. Lys being non-sticky appear to also contribute to relative abundance and amino acid compositions of IDRs (https://doi.org/10.1016/j.jmb.2019.08.008).

6) The size and shape analyses (please see comments of Reviewer 1) need a lot of work and thought. It would help to probe these effects at higher resolution and greater precision.

7) Finally, the asymmetry between interactions that determine single chain dimensions vs. collective phase behavior is puzzling, as noted by Reviewer 2. Please see why this is physically unexpected for simple systems (https://www.sciencedirect.com/science/article/pii/S0006349520304884) and how an asymmetry can arise as discussed recently for prion-like low complexity domains (https://doi.org/10.1038/s41557-021-00840-w). Are the principles uncovered by Bremer et al., operative with the specific IDR studied here? If not, how is the symmetry broken?

*Reviewer #1 (Recommendations for the authors):*

– The manuscript will be strengthened by improving the experimental part. The phase diagrams are not "phase diagrams" in a true sense, they are solubility diagrams or state diagrams of the protein. Phase diagrams are characterized by binodal and tie-lines, which were not measured here. Please see the measured phase diagrams reported in Martin et al. (ref # 18). doi:10.1126/science.aaw8653.

– The use of BF microscopy to distinguish the "morphology" of the condensed phase provides shallow insight into the morphology and dynamics of these assemblies.

– The phase-separated condensates are considered network fluids where percolation and phase separation goes hand-in-hand. A discussion on this should be included in the current manuscript and should include how different variants that the authors studied show distinct percolation behavior. Please see https://doi.org/10.1016/j.molcel.2022.05.018.

– The idea that comes across from reading this manuscript is that IDPs/IDRs are the main driver of phase separation of proteins. This may not be true. It is fine to study an isolated IDR and its phase behavior, but one needs to acknowledge the full-length protein may display a more nuanced behavior through a combination of its IDR and other domains.

– The colloidal cluster formalism, while interesting, should be compared with experimentally determined observables, as the authors point out. Without such data, I am unsure how one can conclude that this formalism provides "a potentially universal foundation" in studying the phase separation of proteins.

– Interactions and solubility of "stickers" and "spacers" have been recently studied by Mittag, Pappu, and co-workers. Such discussions would be helpful to include here since the authors focus on a similar set of residues (R, G, Y, K). Please see https://www.nature.com/articles/s41557-021-00840-w

---

## [Author Response]

Essential revisions:1) Many of the experimental details need careful scrutiny. Both reviewers raise specific concerns and make specific requests. Please respond with all the details that the reviewers are requesting.

Following the suggestions of the Editor and the Reviewers, we have significantly extended the experimental part of the study, including the analysis of sample purity (Figure 1—figure supplement 1A), FRAP-based estimation of condensate dynamic properties (Figure 1D, E and Figure 1—figure supplement 2), analysis of fusion behavior (Videos 1, 2) and circularity analysis of condensate shapes (Figure 1—figure supplement 1C). We have also carried out additional analysis of the previously reported experiments (Figure 1—figure supplement 1A, B and C). Moreover, we have significantly extended the computational/theoretical part of the manuscript, including the analysis of simulation convergence and finite-size effects (Supplementary File 1 “Technical summary”, Figure 4—figure supplement 1C), estimation of translational diffusion coefficients and viscosity (Figures 4C, Figure 4—figure supplement 1D-F, Supplementary File 2) and evaluation of contact probabilities and comparison with percolation theory (Figure 2—figure supplement 1C). Please, find further details below.

2) Both reviewers are concerned about the rather sweeping generalizations and statements made that do not square with the state of the art. First and foremost, it is now clear that the phase transitions in question are not purely segregative LLPS type phenomena. That the field has overused this term and done so without a care is not a good enough reason to perpetuate the false notion of a pure LLPS type behavior. One of the earlier studies demonstrating the coupling of segregative transitions viz., phase separation, and associative transitions, viz., percolation, was published in eLife and elsewhere. Please see: https://elifesciences.org/articles/30294 and http://iopscience.iop.org/article/10.1088/1367-2630/aab8d9. Given the considerable progress made by several labs, and especially those of Mittag and Pappu on the topic of IDR phase transitions, it is imperative that the motivation for the current work not be that LLPS is "poorly understood" (see comments by Reviewer 2).

We thank the Editor and the Reviewers for the detailed, constructive and candid suggestions and criticism. In response, we have extensively revised the manuscript to reflect the current state of knowledge regarding the formation of biomolecular condensates. We fully agree with the Editor that the concept of purely segregative LLPS has been misused in the condensate literature and have paid particular attention in the revision to properly contextualize our work and cite the relevant literature, including the works mentioned by the Editor. Importantly, as further discussed below and in the revised manuscript, the idea of phase separation coupled to percolation (PSCP) provides a natural connection with the notion of fractal scaling, widely observed in different colloidal systems and explored in our manuscript in order to understand the spatial organization of IDP clusters. In combination with percolation theory, the fractal model provides a predictive structural and quantitative perspective on condensate formation and the polydisperse nature of IDRs self-association, as further discussed below. It is precisely this connection between the condensate field and colloidal chemistry that we feel is underexplored and that our work attempts to contribute to. In the revised text, we have articulated these ideas in more detail on pp. 4-5 and 18-19.

3) Please provide a coherent motivation/justification for co-opting concepts such as the fractal analysis from colloidal chemistry for the analysis of the simulations. Please note that this is not the first time fractal analyses have been brought to bear in studying phase separating IDRs. They've been previously deployed in studies of ultra-coarse grained simulations of mimics of the exon 1 encoded region of huntingtin (http://www.sciencedirect.com/science/article/pii/S0006349514007371).

The principal motivation behind our work comes from the known universality of fractal behavior in colloidal systems. Given the close parallels between biomolecular condensates and colloids in many respects, it is natural to critically explore theoretical frameworks that have proven their worth in the latter case. In particular, while fractal scaling has been used to describe the multiscale structure in different colloids, ours is, to the best of our knowledge, the first application of such a formalism to describe the spatial organization of a condensate at an arbitrary scale starting from the atomistic simulations performed on the scale of tens of nanometers. We have discussed these points in the revised manuscript on p. 5. We also thank the Editor for pointing out the above reference, which we now cite and discuss in the revised manuscript. Indeed, the fractal model has been used to interpret the results of coarse-grained simulations and characterize the aggregation of an intrinsically disordered huntingtin fragment. While in the cited article evidence of fractal behavior was found for a subset of scattering vectors, no high-resolution structural details could be provided, primarily because the simulations were coarse-grained. As a complement and an extension of these efforts, our model starts with the atomistic picture and provides a direct prediction of the features of the spatial organization of Lge1_1-80_ condensates at an arbitrary length scale and the impact of different mutations. Of note, the ability of IDRs to form low-dimensional fractal structures upon the disruption of their LLPS tendency by a polyalanine insertion was recently demonstrated experimentally for synthetic elastin-like polypeptides (https://doi.org/10.1038/s41563-018-0182-6), as cited in the revised manuscript.

4) The issues of convergence, statistical robustness, and finite size effects need careful consideration. How do the images interact with one another in the dilute and dense phase simulations. For simple liquids, even the earliest simulations deployed ca. 100 molecules for querying properties of neat liquids. How then does one justify the use of 24 copies for a complex fluid? In lattice-based simulations, the effects of finite size were systematically analyzed, the inference was that one needs at least 100+ molecules to get to coherent descriptions of two coexisting phases. The current work does not simulate coexisting phases but approaches each phase separately. So, fewer molecules are reasonable, but it must still be the case that a reasonably rigorous assessment of finite size effects is provided.

We thank the Editor for this highly relevant comment. Indeed, we did not attempt to capture the process of phase separation or characterize two coexisting phases, for which much larger ensembles would be needed. Rather, our aim was to study the conformational behavior of individual protein chains in the context of a crowded protein mixture, taken as a model for the dense phase, and then use fractal scaling to provide a model of spatial organization of a condensate at an arbitrary length scale. Having said this, it is absolutely important to address how converged the key observables are, given the finite size of the all-atom simulation setup and the limited sampling used. In the revised manuscript, we have included an additional analysis of convergence of our simulations and could show that both key MD-derived parameters required by the fractal model, protein compactness and valency, display convergent behavior over the last 0.3 µs MD in the 24-copy systems (Supplementary File 1, Figure 4—figure supplement 1C). For example, the block averages of compactness and valency exhibit a standard deviation of only 2-4% and 4-8%, respectively, over the last 0.3 µs of MD simulations. Moreover, it should be emphasized that we are interested in single-chain features in the context of a crowded mixture and, thus, our sampling over this range corresponds effectively to 24 x 0.3 µs = 7.2 µs. Finally, a detailed analysis of convergence in conformational sampling was performed for single-copy simulations using calculations of configurational entropy as evaluated by the MIST formalism (Figure 4—figure supplement 1B). Using this measure in the case of the weakly self-interacting Y>A, we indeed do observe a close convergence between two independent replicas over 1-µs trajectories. However, we still recognize the possibility that with longer simulation times and/or more protein copies per simulation, the simulated systems may show a qualitatively different behavior, as discussed on p. 12-13 of the revised manuscript.

Regarding the interaction between simulation images, we should first point out that we employ Ewald summation for the treatment of long-range electrostatics, an approach which by default includes an interaction between every particle and all of its periodic-boundary-condition (PBC) images. We have also directly analyzed direct van-der-Waals contacts between simulation images for each protein in our systems (Supplementary File 1). Importantly, in all 24-copy systems, the average separation between images of protein atoms lies consistently in the 12-15 nm interval and no direct contacts between PBC images are observed. Such large distances also mitigate potential spurious effects due to the usage of Ewald summation. In single-copy WT simulations, the average distance between images in both cases remains above 4 nm and not a single direct contact between images is detected. For the less compact variants (both replicas of Y>A and 1 replica of R>K), the average distance is above 3.6 nm (Supplementary File 1) with a small frequency of transient contacts between the images (less than 0.2 %).

In order to further analyze how realistic our simulations are, we have carried out a detailed analysis of protein translational diffusion and viscosity in the simulations (Figure 4C and Figure 4—figure supplement 1D-F, details in Methods). Our analysis shows signatures of realistic diffusive dynamics in the modeled all-atom systems. Specifically, single-molecule translational diffusion coefficients of Lge1_1-80_ variants obtained from fitting of MSD curves with an applied finite-size PBC correction and solvent viscosity rescaling (see Methods for details) are ~120 µm^2^/s for single-copy simulations or between 100-150 µm^2^/s for 24-copy simulations and different Lge1_1-80_ variants (Figure 4C and Supplementary File 2). This corresponds well to the experimentally measured values for proteins of similar size. For instance, the diffusion constant of the similarly sized ubiquitin (Rg = 1.32 nm, 76 aa, 8.6 kDa) at the concentration of 8.6 mg/ml is 149 µs^2^/s (https://doi.org/10.1038/s41467-021-21181-9), while that of GFP (Rg = 2.8 nm, 238 aa, 27 kDa) at the concentration of 0.5-3 mg/ml is ~90 µs^2^/s (https://doi.org/10.1038/ncomms5494) Interestingly, the obtained viscosity values (see Methods for details) in single-chain simulations of the three Lge1_1-80_ variants (effective concentration of 2.3 mg/ml) are all similar to each other and are close to the reported solvent viscosity for TIP4P-D water/0.1 M NaCl of 0.83 mPa*s (Figure 4—figure supplement 1F and Supplementary File 2). In the crowded 24-copy systems (effective concentration of 6-7 mg/ml), the viscosity systematically increases by about 20 % and is again similar for all three Lge1_1-80_ variants (Figure 4—figure supplement 1F and Supplementary File 2). This reflects the experimental trend that viscosity of protein solutions depends primarily on protein concentration. Importantly, the calculated values correspond well to the experimentally obtained values for e. g. serum albumin in this concentration range (~ 1.1 mPa*s for 10 mg/ml, https://doi.org/10.1039/C5RA21068B).

Note that diffusion coefficients can be more accurately defined in the 24-copy systems than in the single-copy ones. This is due to both, the limited system size and the limitations of MD sampling. For instance, the average diffusion coefficient obtained on the complete MD trajectories for the 24-copy systems relies on 24 µs of MD statistics in total vs. just 2 µs for the single copies. At the same time, finite-size corrections in the case of 24-copy systems are relatively small (35-60 %), while they are almost an order of magnitude higher (450-530 %) for the single-copy ones (Supplementary File 2). Thus, the size-related effects are found to be less dramatic for the modeled 24-copy systems.

5) Regarding the differences between Arg and Lys, the sources of differences are intrinsic and context dependent. As Reviewer 2 notes, this is not as enigmatic as the authors note. Please see recent contributions that have demonstrated clear differences of Arg vs. Lys as drivers of speckle formation (https://doi.org/10.1016/j.molcel.2020.01.025), the realization that Arg and Lys are very different in terms of their intrinsic free energies of hydration (https://pubs.acs.org/doi/abs/10.1021/acs.jpcb.1c01073), and that these differences contribute directly to the cation-specificity of IDP conformational ensembles (https://www.pnas.org/doi/full/10.1073/pnas.2200559119). These physical principles and the distinction of Arg being sticky vs. Lys being non-sticky appear to also contribute to relative abundance and amino acid compositions of IDRs (https://doi.org/10.1016/j.jmb.2019.08.008).

We thank the Editor and the Reviewer 2 for bringing up these studies, which we now cite and discuss in the revised manuscript on pp. 6-8. We should also emphasize that we never intended to claim that the difference between Arg and Lys is generally enigmatic and poorly understood. Namely, the sentence that the Editor and the Reviewer 2 refer to (“*Therefore, the effect of R>K substitution on LLPS should be further explored in the context of protein-protein interactions.”)* was poorly phrased on our part and was only meant in relation to the present study and not in relation to the wider literature on the topic. We simply wanted to refer to the fact that the binding free energies for individual residues do not provide sufficient information about interactions between protein chains. Following the comments of the Editor and Reviewer 2 and to improve clarity, we have rephrased this part and included and discussed additional references (p. 8). Importantly, in agreement with these and other studies (https://doi.org/10.1073/pnas.200022311, https://doi.org/10.1038/s41467-020-18224-y, https://doi.org/10.1038/s41467-022-35001-1, https://doi.org/10.1002/pro.4109), we do see a significant difference in condensate behavior for the R>K mutant in both simulation and experiment. A direct analysis of contact statistics reveals that R-Y is the most dominant type of intermolecular contacts in the crowded mixture of Lge1_1-80_ (see updated statistics in Figure 2—figure supplement 1A and B) and, together with Y-Y, may be the driver of condensate formation in Lge1_1-80_ , in agreement with previous observations by Bremer and coworkers (https://doi.org/10.1038/s41557-021-00840-w). Expectedly, the contribution of Y-Y to protein-protein interactions is substantially increased in crowded mixtures of the R>K mutant. Finally, in contrast to the majority of other studies, which address the general properties of individual amino acids, here we present amino-acid/amino-acid interaction propensities as a function of the polarity of the environment (Figure 1—figure supplement 1D). This complements the recent results by Krainer et al. (https://doi.org/10.1038/s41467-021-21181-9) and provides a quantitative foundation for analyzing the role of cation-pi interactions in condensate formation.

6) The size and shape analyses (please see comments of Reviewer 1) need a lot of work and thought. It would help to probe these effects at higher resolution and greater precision.

Following the comments of the Editor and Reviewer 1, we have purified and fluorescently labelled different constructs (Figure 1—figure supplement 1B) and have used them to characterize and compare different microscopic structures in solution and compare WT vs. R>K and Y>A mutants. Furthermore, we have used circularity analysis to quantify the shape of condensates (Figure 1—figure supplement 1C, details in Methods). In agreement with the predictions of our model, we observe a reduction of the propensity to form condensates in the Y>A mutant. Notably, we have observed amorphous precipitates at high protein concentration of this mutant (45 mM and above), but their material properties (and possible influence of sample impurities) remain unclear. Moreover, due to their sporadic nature, one lacks proper statistics for an adequate quantitative analysis. Hence, we refrain from commenting on these infrequently observed precipitates in the revised manuscript. For further details, please see our replies to the comments of Reviewer 1 below.

7) Finally, the asymmetry between interactions that determine single chain dimensions vs. collective phase behavior is puzzling, as noted by Reviewer 2. Please see why this is physically unexpected for simple systems (https://www.sciencedirect.com/science/article/pii/S0006349520304884) and how an asymmetry can arise as discussed recently for prion-like low complexity domains (https://doi.org/10.1038/s41557-021-00840-w). Are the principles uncovered by Bremer et al., operative with the specific IDR studied here? If not, how is the symmetry broken?

We thank the Editor and the Reviewer 2 for pointing out these relevant studies, which we now cite and discuss. As all constructs in our study have the same net charge, we were not able to analyze the coupling between single-chain and phase behavior with respect to changes in net charge per residue (NCPR) as in the aforementioned studies, and we see this as an important area for future investigation. Having said this, we have compared in detail the sequence composition of Lge1_1-80_ with that of A1-LCD variants studied by Bremer et al. (https://doi.org/10.1038/s41557-021-00840-w) and indeed the principles uncovered by these authors shed light on the asymmetry between single-chain and collective phase behavior of Lge1_1-80_. In particular, when it comes to aromatic composition, Lge1 is most similar to the -12F+12Y mutant of A1-LCD, and by this token, i.e. the high frequency of sticker tyrosines, should exhibit a strong coupling between single-chain and phase behavior. However, the Lge1_1-80_ NCPR of 0.075 is greater than that of A1-LCD (0.059) and this could contribute to the extent of decoupling as suggested by Bremer et al. Moreover, Lge1 is extremely abundant in Arg (13.5 % as compared to 7.4 % in A1-LCD), and in terms of Arg and Lys abundance is most similar to +7R A1-LCD mutant, which showed the greatest degree of decoupling between single-chain and phase behavior in Bremer et al., in agreement with what we see here. While these authors have shown that NCPR is the primary determinant of the extent of decoupling in the case of A1-LCD mutants, including the A1-LCD +7R, their analysis showed that the nature of positive and negative residues involved also makes a significant difference. In particular, the significant excess of Arg residues, as a context-dependent auxiliary sticker, could create the asymmetry between interactions that determine single chain dimensions vs. collective phase behavior.

Furthermore, Martin et al. (https://doi.org/10.1126/science.aaw8653) have shown that an approximately uniform distributions of stickers along the sequence is required for the correspondence between the driving forces behind coil-to-globule transitions and phase separation to hold. We have analyzed the patterning of Tyr residues along the Lge1_1-80_ sequence using W_aro_ parameter used by Martin et al. (note that Tyr is the only aromatic in the Lge1_1-80_ sequence). Interestingly, W_aro_ of the native Lge1 sequence (0.47) falls in the middle of the distribution for its shuffled variants (p=0.57), in contrast to the highly patterned sequences such as that of A1-LCD with p>0.99. Taken together, in addition to NCPR, symmetry breaking in the case of Lge1_1-80_ could be a consequence of its complex sequence composition, including both the non-uniform patterning of tyrosines and a high abundance of arginines. Provided that our simulations are long enough to provide an equilibrium picture and are on the length-scale of a single protein not strongly influenced by finite-size effects (these potential artifacts cannot be discounted), they actually can be seen as a demonstration of such symmetry breaking in a heteropolymer, as Reviewer 2 accepts. The above comparisons and discussion, while extremely important, are outside the scope of the present manuscript and will be treated in a separate manuscript.

Furthermore, analysis of pairwise contacts suggests that intra- and intermolecular interactions rely on a similar pool of contacts by amino-acid type, but differ significantly if one analyzes specific sequence location of the interacting residues involved (Figure 2—figure supplement 1A and B). For example, one observes a high correlation between the frequencies of different contacts by amino-acid type when comparing intramolecular contacts in single-copy simulations and intermolecular contacts in 24-copy simulations (Figure 3—figure supplement 1B). This correlation is completely lost (Figure 3—figure supplement 1C) if one analyzes position-resolved statistics (2D pairwise contacts maps) or statistically defined interaction modes (Figure 3A, and Figure 3—figure supplement 1A). For example, although Tyr-Tyr interactions dominate in both cases, in single-copy simulations of WT Lge1_1-80_ the C-terminal Tyr_80_ barely participates in any intramolecular interactions with other residues (Figure 3—figure supplement 1A), while in 24-copy simulations it is one of the most intermolecularly interactive residues (Figure 3A). In other words, while the symmetry between intra- and intermolecular interactions can be observed at the level of pairwise contact types (similar type contact used for both), the distribution of these contacts along the peptide sequence is clearly different in the two cases. Finally, it should be mentioned that the parallel between single-copy and phase behavior in both homopolymers and heteropolymers is observed primarily at the level of thermodynamic variables such as LLPS critical temperature (Tc), coil-to-globule transition temperature (T_q_) or the Boyle temperature (T_B_). It is possible that the noted correspondence extends primarily to such and similar thermodynamic variables, while more structural, topological features of the globule in the single-molecule case and the network in the collective phase case remain uncoupled.

Interestingly, the core of intramolecular interactions observed for a single molecule at infinite dilution and in the crowded context remain approximately the same as reflected in the high correlation between intramolecular modes obtained in single and multichain simulations. Namely, proteins keep core self-contacts and establish new ones with neighbors, but do not lose “self-identity”, as in homopolymer melts. Similar effects have also been observed elsewhere: https://doi.org/10.1073/pnas.2000223117, https://doi.org/10.1073/pnas.1804177115.

Reviewer #1 (Recommendations for the authors):– The manuscript will be strengthened by improving the experimental part. The phase diagrams are not "phase diagrams" in a true sense, they are solubility diagrams or state diagrams of the protein. Phase diagrams are characterized by binodal and tie-lines, which were not measured here. Please see the measured phase diagrams reported in Martin et al. (ref # 18). doi:10.1126/science.aaw8653.

We indeed could not measure complete binodal and tie-lines, due in part to the experimental limitations when it comes to working with high protein concentrations for Lge1_1-80_ constructs. Therefore, as suggested by the Reviewer, we now refer to the presented diagrams as “solubility diagrams”.

– The use of BF microscopy to distinguish the "morphology" of the condensed phase provides shallow insight into the morphology and dynamics of these assemblies.

Following the Reviewer’s comment, we have purified and fluorescently labelled different constructs. This has allowed us to use fluorescence microscopy for a more detailed analysis of the condensed phase (Figure 1—figure supplement 1B. Please, see also above). Different morphologies have been assessed by quantifying the circularity of the observed condensates (Figure 1—figure supplement 1C). Moreover, we have included the analysis of the dynamics of these assemblies at different levels by fusion experiments (Video 1 and Video 2) and FRAP (Figure 1D, E and Figure 1—figure supplement 2). Given the fact that protein precipitates of the Y>A mutant are present at extremely low abundance and only at concentrations of 45 µM and above, fusion behavior could not be studied in that case.

– The phase-separated condensates are considered network fluids where percolation and phase separation goes hand-in-hand. A discussion on this should be included in the current manuscript and should include how different variants that the authors studied show distinct percolation behavior. Please see https://doi.org/10.1016/j.molcel.2022.05.018.

Following the Reviewer’s suggestion, we have thoroughly revised the manuscript in order to connect and contrast our findings with the percolation/phase separation framework. Indeed, the fractal behavior is naturally related to percolation phenomena. According to the definition used in colloidal chemistry, biological condensates can be described as a weak gel, which undergoes a transition between a population of finite-size clusters (‘pre-percolation clusters’) or sol, and an infinitely large cluster or gel (https://doi.org/10.1007/3-540-11471-8_4). In the case of biological condensates, phase separation coupled to percolation results in finite-sized droplets and the appearance of surface tension. Under the requirement that c_sat_ < c_perc_ < c_dense_, phase separation leads to an increase in local concentration of IDPs and defines the phase boundary, while percolation transition establishes the network connectivity (https://doi.org/10.1016/j.molcel.2022.05.018). Consequently, the spatial organization of both finite-sized and infinite clusters is directly connected to a networking transition (percolation). For instance, direct links between the fractal dimension defining scaling principles in self-similar clusters and critical exponents in different percolation models have been provided (https://doi.org/10.1007/3-540-11471-8_4, https://doi.org/10.1007/BF01012940).

While a similar formal derivation for our model is out of the scope of the present study, we can provide an illustration of how our results and the key parameters of the model can be interpreted according to percolation theory. A key concept in percolation theory is that of contact probability between components in the system. The network transition appears, and a percolating cluster is formed if the contact probability exceeds a particular threshold i.e. the critical contact probability (*p_crit_*). For instance, in the simplest case according to the Flory-Stockmayer theory *p_crit_* = 1/(n-1), where n is the number of bonds formed by each monomer and is related to the valency in our model. Thus, interaction valency contributes directly to the spatial organization of pre-percolating clusters (the fractal dimension) and defines the threshold of percolation. In the revised manuscript, we have estimated the contact probabilities (*p_c_*) for all three Lge1_1-80_ variants, under the assumption of a well-mixed system, that is, under the assumption that all chains in the simulation box can, in principle, establish contacts with all the other chains. As shown in Figure 2—figure supplement 1C, contact probabilities evolve in direct proportion to valency, with a plateau over the last 0.3 µs. Importantly, the WT contact probability reaches a level that is ~1.5 fold higher than for either mutant. While we cannot independently estimate the value of *p_c_* in our simulated systems, the fact that on the simulation time scale WT forms a single percolating cluster and the two mutants do not (Figure 2C), is consistent with this difference in contact probability. We have highlighted these points on p. 19 of the revised manuscript.

– The idea that comes across from reading this manuscript is that IDPs/IDRs are the main driver of phase separation of proteins. This may not be true. It is fine to study an isolated IDR and its phase behavior, but one needs to acknowledge the full-length protein may display a more nuanced behavior through a combination of its IDR and other domains.

We fully agree with the Reviewer that IDPs/IDRs are not the only drivers of protein phase separation and that folded domains play a critical role in many systems. We have emphasized this point on pp. 5-6, 20 of the revised manuscript. Stimulated by the Reviewer’s comment, we have also realized that the role of the Lge1 IDR in driving phase separation merits a clearer explanation. In response, we now provide a more extensive discussion of the contribution of different domains of Lge1 to both its phase behavior and biological function (pp. 5-6, 20). Specifically, in a previous study by Gallego et al. (https://doi.org/10.1038/s41586-020-2097-z), it was shown that the first 80 aa of Lge 1 are the main driver of its phase separation, with the rest of the IDR also contributing to phase separating properties. Moreover, the other key domain of Lge1 – its C-terminal predicted coiled coil – has an essential function in vivo due to its interaction with the E3 ligase Bre1, which is required for histone H2B ubiquitination. We should emphasize that in the present manuscript, the Lge1_1-80_ fragments and its mutants were used as a model system to study the physicochemical features of the respective condensates/precipitates, with the biological interpretations being out of the scope of the present study.

– The colloidal cluster formalism, while interesting, should be compared with experimentally determined observables, as the authors point out. Without such data, I am unsure how one can conclude that this formalism provides "a potentially universal foundation" in studying the phase separation of proteins.

A key contribution of the present work is the development and the definition of a quantitative model that treats the spatial organization of a biomolecular condensate across scales using just two key parameters that capture the behavior of individual polymer chains in the condensate (valency and compactness). As the Reviewer correctly points out, the extensive quantitative predictions of the model should be thoroughly tested against equally detailed experimental data. In the present manuscript, we have taken the first, largely qualitative steps in this direction. Specifically, the model can be used to reconstruct the spatial organization of clusters of arbitrary size at the atomistic level (Figure 5A and B, Videos 4, 5, and 6), enabling a structural understanding of the organization of condensate interior. A direct application of such understanding concerns the nature of cavity sizes and interpretation of dextran partitioning experiments. Second, as pointed above, differences in morphology of protein clusters propagate across scales, and can be qualitatively characterized by the analysis of microscopic images i.e. presence or absence of detectable condensates as a function of the fractal dimension *d_F_* (see also discussion above). In particular, the model correctly predicts the difference in the behavior of WT and R>K as opposed to Y>A variants, solely based on the predicted fractal dimension they exhibit. Finally, we could show that the MD simulations indeed match the predictions of fractal scaling for the three smallest clusters. Here, it is important to understand that MD simulations in the first instance just give the average valency and compactness of individual chains in the dense phase. These values are then input into the fractal scaling formalism, which is conceptually fully independent from MD simulations, to obtain the dependence of condensate mass on its radius, M(R), at any desired length scale. The analysis presented in Figure 5—figure supplement 1B and discussed on p. 16 shows that the predictions of fractal scaling for the first three smallest clusters indeed correspond to what is seen in MD. This is a non-trivial correspondence and can be taken as direct evidence that fractal organization is present even at the shortest scale, i.e. at the level of MD simulation boxes. Ultimately, static light scattering experiments would give the best possibility to test the model directly. However, these experiments are beyond the scope of the present work, concerned with presenting the quantitative model and linking it with MD simulations, and will be the topic of our future work. In particular, the fractal formalism predicts significant regions of linear behavior in such curves in log-log representation, while the fractal dimension *d*_F_, provides a quantitative point of comparison between theoretical predictions and experimental measurements. Following the Reviewer’s comment, we have rephrased the above statement to better reflect these points (p. 22).

– Interactions and solubility of "stickers" and "spacers" have been recently studied by Mittag, Pappu, and co-workers. Such discussions would be helpful to include here since the authors focus on a similar set of residues (R, G, Y, K). Please see https://www.nature.com/articles/s41557-021-00840-w

Indeed, a number of findings in our study are in line with Bremer et al. (https://doi.org/10.1038/s41557-021-00840-w), as discussed above in our reply to the Editor’s comments. As already pointed out, this concerns both similar trends in amino-acid interaction preferences, whereby R-Y and Y-Y are the drivers of IDR self-association, while R>K mutations modulate the LLPS potential, as well as symmetry breaking between intra- and intermolecular interaction modes. We have included the corresponding discussion in the revised manuscript on pp. 8-9, 11.